# Spectral Convolutional Conditional Neural Processes

**Peiman Mohseni**
Texas A&M University
`peiman.mohseni@tamu.edu`

**Nick Duffield**
Texas A&M University
`duffieldng@tamu.edu`

## Abstract

Neural Processes (NPs) are meta-learning models that learn to map sets of observations to approximations of the corresponding posterior predictive distributions. By accommodating variable-sized, unstructured collections of observations and enabling probabilistic predictions at arbitrary query points, NPs provide a flexible framework for modeling functions over continuous domains. Since their introduction, numerous variants have emerged; however, early formulations shared a fundamental limitation: they compressed the observed data into finite-dimensional global representations via aggregation operations such as mean pooling. This strategy induces an intrinsic mismatch with the infinite-dimensional nature of the stochastic processes that NPs intend to model. Convolutional conditional neural processes (ConvCNPs) address this limitation by constructing infinite-dimensional functional embeddings processed through convolutional neural networks (CNNs) to enforce translation equivariance. Yet CNNs with local spatial kernels struggle to capture long-range dependencies without resorting to large kernels, which impose significant computational costs. To overcome this limitation, we propose spectral ConvCNPs (SConvCNPs), which perform global convolution in the frequency domain. Inspired by Fourier neural operators (FNOs) for learning solution operators of partial differential equations (PDEs), our approach directly parameterizes convolution kernels in the frequency domain, leveraging the relatively compact yet global Fourier representation of many natural signals. We validate the effectiveness of SConvCNPs on both synthetic and real-world datasets, demonstrating how ideas from operator learning can advance the capabilities of NPs.

## 1 Introduction

Stochastic processes offer a mathematical framework for modeling systems that evolve with inherent randomness over continuous domains such as time and space. They underpin a wide range of scientific applications—from spatio-temporal climate dynamics to biological and physical systems—thereby motivating the development of machine learning methods that can learn from data generated by such phenomena [Mathieu et al., 2021, Vaughan et al., 2021, Allen et al., 2025, Ashman et al., 2025, Dupont et al., 2021]. Among classical approaches, Gaussian processes (GPs; Rasmussen et al., 2006) provide a Bayesian framework with closed-form inference and uncertainty quantification. However, their cubic computational cost from matrix inversion and the difficulty of specifying suitable kernels—especially in high-dimensional settings—limit their scalability.

Motivated by the success of deep neural networks in large-scale function approximation, neural network-based alternatives have emerged. Neural processes (NPs; Garnelo et al., 2018a,b) exemplify this paradigm, combining ideas from GPs and deep learning within a meta-learning framework. By exposing the model to multiple realizations of an underlying stochastic process, each treated as a distinct task, NPs learn shared structures across tasks to parameterize a neural mapping that *directly* gives an approximation of the corresponding posterior predictive distribution [Bruinsma, 2024]. Once trained, the model enables efficient probabilistic predictions on new tasks without further training.

39th Conference on Neural Information Processing Systems (NeurIPS 2025).

Since the introduction of conditional neural processes (CNPs, Garnelo et al. [2018a]) as the first class within the NPs family [Jha et al., 2022], numerous extensions have been proposed to enhance their effectiveness. One prominent line of work focuses on incorporating explicit inductive biases into CNPs in order to better capture the symmetries that commonly arise in scientific applications [Gordon et al., 2019, Kawano et al., 2021, Holderrieth et al., 2021, Huang et al., 2023, Ashman et al., 2024a,b]. Another major direction seeks to move beyond the mean-field factorized Gaussian predictive distributions to which CNPs are limited. A widely adopted approach augments CNPs with stochastic latent variables, giving rise to the family of latent neural processes (LNPs; Garnelo et al. [2018b], Louizos et al. [2019], Wang and Van Hoof [2020], Foong et al. [2020], Lee et al. [2020], Volpp et al. [2021], Wang et al. [2022], Wang and van Hoof [2022], Kim et al. [2022], Jung and Park [2023], Lee et al. [2023], Xu et al. [2023]). Complementary efforts explore autoregressive prediction schemes [Bruinsma et al., 2023, Nguyen and Grover, 2022], Gaussian predictive distributions with non-diagonal covariances [Bruinsma et al., 2021, Markou et al., 2022], and quantile-based parameterizations of the predictive distribution [Mohseni et al., 2023].

This work focuses on CNPs, particularly convolutional CNPs (ConvCNPs; Gordon et al. [2019]), which were the first to endow NPs with translation equivariance. ConvCNPs introduce the convolutional deep set construction which characterizes a broad class of translation-equivariant mappings over finite, potentially unstructured sets of observations as a composition of a functional embedding with a translation-equivariant operator, typically realized through convolutional neural networks (CNNs; Fukushima [1980], LeCun et al. [1989, 1998]).

Despite their effectiveness, ConvCNPs can struggle to aggregate information from observations spread across large spatial domains—a challenge that becomes particularly pronounced in sparse data regimes. This limitation stems from their reliance on local convolutional kernels with small receptive fields, which hampers their ability to model long-range dependencies. A natural remedy is to enlarge the kernel size to extend the receptive field; however, this approach rapidly increases the number of model parameters and computational cost [Romero et al., 2021]. Alternatively, transformer-based architectures can capture long-range interactions but incur quadratic, rather than linear, computational complexity in the number of observations [Vaswani et al., 2017, Nguyen and Grover, 2022].

In this work, we pursue an alternative paradigm that represents functions in the frequency domain, inspired by the well-established observation that many natural processes exhibit energy concentration in low-frequency bands [Field, 1987, Ruderman and Bialek, 1993, Wainwright and Simoncelli, 1999]. This property allows for efficient approximation using only a subset of dominant spectral coefficients, enabling tractable computations while preserving the signal's global structure. By parameterizing convolution kernels directly in the Fourier domain over a finite set of frequencies and leveraging the convolution theorem, we can attain large effective receptive fields without incurring prohibitive computational costs.

While spectral methods have been extensively studied in neural operator learning for partial differential equations (PDEs, Chen and Chen [1995], Kovachki et al. [2021, 2023]), their application within NPs framework remains relatively unexplored. To bridge this gap, we propose spectral convolutional conditional neural processes (SConvCNPs)—a class of models that adopt Fourier neural operators (FNOs; Li et al. [2020a]) to realize global spectral convolution while maintaining computational efficiency. Across a suite of synthetic and real-world benchmarks, SConvCNPs perform competitively with state-of-the-art baselines, illustrating how ideas from neural operators can enhance the flexibility and performance of NPs.

## 2 Preliminaries

### 2.1 Fourier Neural Operators

Neural operators [Chen and Chen, 1995, Li et al., 2020a, Kovachki et al., 2023] are neural network architectures designed to learn mappings between *function spaces* rather than finite-dimensional vectors. Like conventional feed-forward networks, they consist of stacked layers that alternate between operator-based transformations and pointwise nonlinearities. A common transformation is an integral operator with kernel $\kappa : \mathcal{X} \times \mathcal{X} \to \mathcal{Y}$, acting on an input function $v : \mathcal{X} \to \mathcal{Y}$ as

$$\mathcal{K}[v](x) = \int \kappa(x, s)\, v(s)\, \mathrm{d}s,$$

where $\mathcal{X} = \mathbb{R}^{d_x}$ and $\mathcal{Y} = \mathbb{R}^{d_y}$ are Euclidean spaces with $d_x, d_y \in \mathbb{N}$. While this work focuses on operators linear in $v$, nonlinear formulations have also been explored, including continuous formulations of softmax attention [Ashman et al., 2024a, Calvello et al., 2024]. When the kernel is *stationary*—that is, $\kappa(x, s) = \kappa(x - s)$—the operator reduces to a convolution, $\mathcal{K}[v] = \kappa * v$, connecting neural operators to CNNs where $\kappa$ is parameterized by learnable weights.

Convolutional kernels are typically spatially local with limited receptive fields [Luo et al., 2016, Peng et al., 2017, Wang et al., 2018]. Modeling long-range dependencies thus requires large kernels, substantially increasing parameter count. The Fourier Neural Operator (FNO; Li et al. [2020a]) addresses this limitation by exploiting the convolution theorem [Bracewell and Kahn, 1966, Oppenheim, 1999], which expresses convolution as

$$\mathcal{K}[v](x) = \mathcal{F}^{-1}\left[\widehat{\kappa}(\xi) \cdot \widehat{v}(\xi)\right](x), \tag{1}$$

where $\widehat{f} := \mathcal{F}[f]$ denotes the Fourier transform of $f$, and $\mathcal{F}^{-1}$ denotes the inverse Fourier transform. Rather than parameterizing $\kappa$ in the spatial domain, the FNO learns its Fourier representation $\widehat{\kappa}$ directly. When $v$ is approximately band-limited—i.e., $\widehat{v}(\xi)$ has negligible energy for $\|\xi\| > \xi_0$—high-frequency components can be truncated with minimal information loss. This property, observed in many natural signals [Field, 1987, Ruderman and Bialek, 1993, Wainwright and Simoncelli, 1999], allows setting $\widehat{\kappa}(\xi) = 0$ outside the retained band without significant degradation.

In practice, functions are accessible only through discrete samples, requiring the discrete Fourier transform (DFT) for domain transitions. Given samples of $v$ on a *uniform* grid $\mathcal{G} \subset \mathcal{X}$, the FNO applies the DFT via the fast Fourier transform (FFT; Cooley and Tukey [1965], Frigo and Johnson [2005]). The resulting spectrum is truncated to a finite set of frequency modes $\Xi \subset \mathbb{R}^{d_x}$ assumed to capture most of the signal's energy. For each retained frequency $\xi \in \Xi$, the Fourier kernel is parameterized by learnable complex weights $W_\xi \in \mathbb{C}^{d_y}$, so that $\widehat{\kappa}(\xi) = W_\xi$.[1] This formulation implicitly imposes spatial periodicity, as the kernel is represented using discrete harmonics (equivalently, a Dirac comb in frequency space). After pointwise multiplication in frequency space, an inverse FFT returns the operator output to the spatial domain.

## 2.2 Neural Processes

Let $\mathcal{P}(\mathcal{Y}^{\mathcal{X}})$ denote the space of probability measures over measurable functions $f : \mathcal{X} \to \mathcal{Y}$, which we interpret as stochastic processes indexed by $\mathcal{X}$. We assume an unknown data-generating process $\mu \in \mathcal{P}(\mathcal{Y}^{\mathcal{X}})$, from which latent functions are drawn. A *task* consists of a *finite* number of noisy input–output observations generated from a realization $f \sim \mu$, partitioned into a context set and a query set:

$$\mathcal{D} = (\mathcal{D}_c, \mathcal{D}_q), \qquad \mathcal{D}_c = \{(x_{c,k}, y_{c,k})\}_{k=1}^{n_c}, \qquad \mathcal{D}_q = \{(x_{q,l}, y_{q,l})\}_{l=1}^{n_q}. \tag{2}$$

Observations are generated according to

$$y_{c,k} = f(x_{c,k}) + \epsilon_{c,k}, \qquad y_{q,l} = f(x_{q,l}) + \epsilon_{q,l},$$

where $\epsilon_{c,k}, \epsilon_{q,l} \overset{\text{i.i.d.}}{\sim} \mathcal{N}(0, \sigma_0^2)$ and $\sigma_0 > 0$ denotes the observation noise standard deviation.

Neural processes (NPs; [Garnelo et al., 2018a,b]) constitute a class of models that use neural networks to learn a mapping $\eta : \mathcal{S}(\mathcal{X} \times \mathcal{Y}) \to \mathcal{P}(\mathcal{Y}^{\mathcal{X}})$, where $\mathcal{S}(\mathcal{X} \times \mathcal{Y})$ denotes the collection of all *finite* subsets of $\mathcal{X} \times \mathcal{Y}$. Given a context set $\mathcal{D}_c$, $\eta$ outputs a stochastic process intended to approximate the Bayesian posterior over functions induced by $\mu$ and conditioned on $\mathcal{D}_c$. Typically, this process is specified *implicitly* through its finite-dimensional marginals [Garnelo et al., 2018b, Bruinsma et al., 2021, Bruinsma, 2024, Mathieu et al., 2023]. Let $\mathbf{x}_q = (x_{q,l})_{l=1}^{n_q}$ and $\mathbf{y}_q = (y_{q,l})_{l=1}^{n_q}$. We denote by $\eta[\mathcal{D}_c; \mathbf{x}_q](\cdot)$ the finite-dimensional distribution of the process $\eta[\mathcal{D}_c]$ evaluated at $\mathbf{x}_q$, and by $\mu_{\mathbf{x}_q}(\cdot \mid \mathcal{D}_c)$ the corresponding finite-dimensional marginal of the true posterior process. Informally, the NP approximation aims to satisfy

$$\eta[\mathcal{D}_c; \mathbf{x}_q](\cdot) \approx \mu_{\mathbf{x}_q}(\cdot \mid \mathcal{D}_c),$$

for all $\mathbf{x}_q$ and $\mathcal{D}_c$ [Bruinsma et al., 2021, Bruinsma, 2024]. With slight abuse of notation, we write $\eta[\mathcal{D}_c; \mathbf{x}_q](\mathbf{y}_q)$ for the density of this finite-dimensional distribution evaluated at $\mathbf{y}_q$.

---

[1]More generally, a matrix-valued parameterization $W_\xi \in \mathbb{C}^{c_{\text{out}} \times d_y}$ is used, where $c_{\text{out}}$ is the number of output channels, enabling joint mixing of input channels and projection to different dimensionality.

In this work, we focus on *conditional neural processes* (CNPs; Garnelo et al. [2018a]), which restrict these finite-dimensional distributions to mean-field Gaussians, i.e. $\eta[\mathcal{D}_c; \mathbf{x}_q] = \prod_{l=1}^{n_q} \eta[\mathcal{D}_c; x_{q,l}]$, where each marginal $\eta[\mathcal{D}_c; x_{q,l}]$ is Gaussian. Each predictive marginal is typically parameterized via a two-stage encoder–decoder architecture [Bruinsma, 2024, Ashman et al., 2024a, 2025]. The encoder $\varphi_e : \mathcal{S}(\mathcal{X} \times \mathcal{Y}) \to \mathcal{H}$ maps the context set $\mathcal{D}_c$ to a latent representation, while the decoder $\varphi_d : \mathcal{H} \to \Theta^{\mathcal{X}}$ maps this representation to a function that assigns, to each query $x_{q,l}$, parameters $\theta(x_{q,l}) \in \Theta$ of the predictive distribution $\eta[\mathcal{D}_c; x_{q,l}]$.

The vanilla CNP encoder summarizes $\mathcal{D}_c$ using a *permutation-invariant* architecture [Qi et al., 2017, Zaheer et al., 2017]. Each context pair $(x_{c,k}, y_{c,k}) \in \mathcal{D}_c$ is independently mapped by a shared network to a finite-dimensional embedding $\varepsilon_{c,k}$, which are then aggregated—typically via mean pooling—into a single representation $\varepsilon_c$. Notably, this encoding is independent of the query locations. The decoder then combines $\varepsilon_c$ with each $x_{q,l}$ to parameterize the Gaussian predictive distribution.

Although sum-pooling aggregation provides universal approximation guarantees [Zaheer et al., 2017, Bloem-Reddy and Teh, 2020], NPs employing such mechanisms often exhibit underfitting in practice [Kim et al., 2019]. Prior works have partly attributed this phenomena to two primary factors [Xu et al., 2020]: (1) the limitation of summaries with *prespecified finite dimensionality* in representing context sets of arbitrary size [Wagstaff et al., 2019], and (2) the shortcomings of simple sum or mean pooling operations to effectively capture rich dependencies between different points [Xu et al., 2020, Nguyen and Grover, 2022].

Since NPs address inherently functional learning problems, it is natural to consider embeddings that themselves take the form of functions. Gordon et al. [2019] introduce a framework for translation-equivariant prediction maps over sets, satisfying

$$\eta\Big[\{(x_c + \tau, y_c) \mid (x_c, y_c) \in \mathcal{D}_c\}; \mathbf{x}_q\Big] = \eta\big[\mathcal{D}_c; \mathbf{x}_q - \tau\big],$$

for all translations $\tau$, context sets $\mathcal{D}_c$, and collections of query locations $\mathbf{x}_q$, where subtraction by $\tau$ acts pointwise, i.e., $\mathbf{x}_q - \tau = (x_{q,l} - \tau)_{l=1}^{n_q}$. They show that a broad class of such maps can be written as $\eta[\mathcal{D}_c] = \varphi_d \circ \varphi_e[\mathcal{D}_c]$, where the functional embedding is defined by

$$\varphi_e[\mathcal{D}_c](x) = \sum_{(x_c, y_c) \in \mathcal{D}_c} \phi(y_c)\, \psi_e(x - x_c). \tag{3}$$

Moreover, $\varphi_d : \mathcal{H} \to C_b(\mathcal{X}, \mathcal{Y})$ is a translation-equivariant decoder acting on a function space $\mathcal{H}$, $C_b(\mathcal{X}, \mathcal{Y})$ denotes the space of bounded continuous functions from $\mathcal{X}$ to $\mathcal{Y}$, $\phi(y) = (1, y)^2$, and $\psi_e : \mathcal{X} \to \mathbb{R}$ is a continuous strictly positive-definite kernel, typically Gaussian.

In convolutional conditional neural processes (ConvCNPs), the functional embedding $\varphi_e[\mathcal{D}_c]$ is evaluated on a uniform grid $\mathcal{G} \subset \mathcal{X}$ covering the joint support of $\mathbf{x}_c$ and $\mathbf{x}_q$, yielding the discretized representation $(\varphi_e[\mathcal{D}_c](x_g))_{x_g \in \mathcal{G}}$. This representation is processed by $\varphi_d$, and predictions at query locations are obtained via kernel interpolation:

$$\theta_{q,l} = \sum_{x_g \in \mathcal{G}} (\varphi_d \circ \varphi_e)[\mathcal{D}_c](x_g)\, \psi_d(x_{q,l} - x_g), \qquad l = 1, \ldots, n_q, \tag{4}$$

where $\psi_d$ is another strictly positive-definite kernel. This interpolation step may be viewed as part of the decoder, preserving the overall encoder–decoder abstraction.

## 3 Spectral Convolutional Conditional Neural Processes

The decoder $\varphi_d$ in ConvCNPs is typically parameterized using standard CNNs such as U-Net [Ronneberger et al., 2015] or ResNet [He et al., 2016]. These architectures employ discrete convolutional kernels—finite sets of learnable parameters that define localized filters operating over neighboring grid points. The kernel size, fixed *a priori*, determines the receptive field of each convolution [Ding et al., 2022] and is generally much smaller than the overall extent of the input signals in the physical domain [Romero et al., 2021, Knigge et al., 2023]. This locality constraint fundamentally limits a model's ability to capture long-range dependencies and to integrate information from observations

---

[2]More generally, $\phi(y) = (1, y, \ldots, y^M)$, where $M$ accounts for repeated inputs. See Gordon et al. [2019] and Bruinsma [2024] for details.

distributed across large spatial or temporal domains [Peng et al., 2017, Wang et al., 2018, Ramachandran et al., 2019, Wang et al., 2020]. The issue becomes particularly pronounced when handling sparse or irregularly sampled data, where effective global reasoning cannot emerge solely from local convolution operations. Although enlarging the convolutional kernel increases the receptive field, it results in a rapid escalation of both parameter count and computational cost. Transformer-based architectures mitigate this issue by enabling explicit global interactions [Vaswani et al., 2017]; however, they incur quadratic, rather than linear, complexity in the number of inputs, rendering them impractical for large context sets unless one resorts to approximation schemes [Nguyen and Grover, 2022, Feng et al., 2022, Ashman et al., 2024a, 2025].

To overcome this limitation without relying on prohibitively large filters or expensive transformers, we exploit the Fourier representation of signals. This is motivated by the well-established observation that many natural signals are approximately band-limited (see Section 2.1), implying that their Fourier representation offers a more compact encoding *relative* to its physical-domain counterpart. Concretely, we instantiate $\varphi_d$ via spectral convolution modules based on equation 1. This substitution effectively enlarges the receptive field, enabling the model to capture global structures from sparse or irregularly sampled data without incurring a parameter count explosion. We refer to the resulting models as spectral convolutional conditional neural processes (SConvCNPs).

**Computational Complexity.** The computational cost of SConvCNPs comprises three parts: (i) $\mathcal{O}(|\mathcal{D}_c||\mathcal{G}|)$ to compute the discretized functional embedding on the grid $\mathcal{G}$ (equation 3); (ii) $\mathcal{O}(|\mathcal{G}| \log |\mathcal{G}|)$ for the FFT-based spectral convolutions (equation 1); and (iii) $\mathcal{O}(|\mathcal{D}_q||\mathcal{G}|)$ to interpolate grid embeddings at the query locations (equation 4). Overall, the total complexity is $\mathcal{O}\big(|\mathcal{G}|(|\mathcal{D}_c| + \log |\mathcal{G}| + |\mathcal{D}_q|)\big)$, matching the $\mathcal{O}\big(|\mathcal{G}|(|\mathcal{D}_c| + 1 + |\mathcal{D}_q|)\big)$ complexity of ConvCNPs up to logarithmic factors. Both architectures therefore scale *linearly* with task size. A key limitation, however, is that $|\mathcal{G}|$ grows exponentially with the input dimension $d_x$, restricting these methods to low-dimensional domains. In contrast, transformer-based NPs (TNPs; Kim et al. [2019], Nguyen and Grover [2022], Feng et al. [2022], Ashman et al. [2024a]) avoid gridding entirely and thus scale more gracefully with input dimensionality. Instead, their computational cost scales quadratically with the task size, typically as $\mathcal{O}(|\mathcal{D}_c|^2 + |\mathcal{D}_q|^2)$, with minor variations depending on the specific implementation (see Section B.1.1).

**Positional Encodings** The convolution operator preserves translation equivariance under the Fourier transform (see equation 1). However, practical FNOs implementations often include explicit positional information to improve predictive accuracy [Li et al., 2020a, Tran et al., 2021, Gupta et al., 2021, Rahman et al., 2022b, Helwig et al., 2023, Tripura and Chakraborty, 2023, Liu et al., 2023a, Li et al., 2024]. Accordingly, we augment the functional embedding $\varphi_e[\mathcal{D}_c](x)$ with positional information:

$$\widetilde{\varphi}_e[\mathcal{D}_c](x) = \big(\varphi_e[\mathcal{D}_c](x),\, x\big).$$

While this augmentation explicitly breaks translation equivariance, we empirically observe performance improvements (see Section B.5). An interesting direction for future work is to investigate *relative* positional encodings [Shaw et al., 2018, Su et al., 2024], which can provide spatial context while preserving translation equivariance.

**Discretization Sensitivity of DFT.** Unlike the continuous Fourier transform, the DFT—and therefore the FFT—depends inherently on the grid $\mathcal{G}$ on which $\varphi_e[\mathcal{D}_c]$ (or $\widetilde{\varphi}_e[\mathcal{D}_c]$) is represented. This dependence arises from both the grid resolution and its physical extent: changing either alters the resulting Fourier coefficients and can lead to inconsistent behavior in the outputs (see Section A). Sensitivity to resolution is not unique to DFT; CNNs exhibit analogous issues [Raonic et al., 2023, Bartolucci et al., 2023]. For example, ConvCNPs mitigate this effect by fixing the grid resolution. While spatial CNNs become stable once the resolution is fixed, DFT-based methods remain sensitive unless *both* the resolution and the physical range are controlled. Accordingly, we fix both, choosing a domain sufficiently large to cover all context and query inputs across tasks. When this is impractical, the domain can instead be divided into (possibly overlapping) fixed-size patches, with spectral convolutions applied independently to each patch and the outputs aggregated—mirroring the mechanism of standard convolution layers. Related ideas in efficient transformer architectures suggest that this is a promising direction for future work [Beltagy et al., 2020, Zaheer et al., 2020, Liu et al., 2021, Ding et al., 2023].

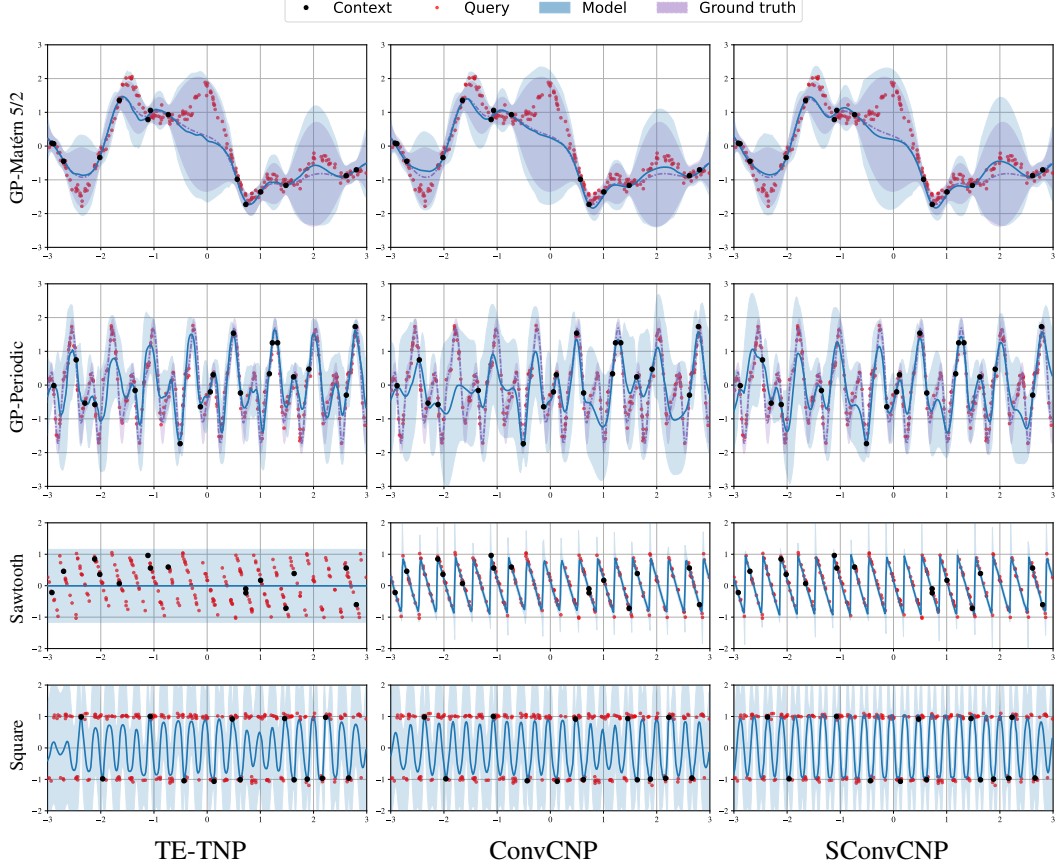

Figure 1: Example predictions on synthetic data. For each model, the blue curve denotes the predictive mean, and the shaded region corresponds to $\pm 2$ standard deviations of the model's Gaussian predictive distribution. In the first two rows, where the data are generated from Gaussian processes, the ground-truth distribution is shown in purple: the dash–dotted curve indicates the true mean, and the shaded band represents $\pm 2$ standard deviations around it. Black points denote context observations, and the red point indicates the queries.

## 4 Experiments

We evaluate our framework on four regression benchmarks and compare its performance against several members of CNPs family. Specifically, we include the original CNP [Garnelo et al., 2018a], the Attentive CNP (AttCNP; Kim et al. [2019]), the Convolutional CNP (ConvCNP; Gordon et al. [2019]), the diagonal variant of the Transformer Neural Process (TNP; Nguyen and Grover [2022]), and the Translation-Equivariant Transformer Neural Process (TE-TNP; Ashman et al. [2024a]). For each experimental setting, all models are trained using four random seeds. Performance is evaluated using the final aggregated log-likelihood and root-mean-squared error (RMSE), and we report the mean $\pm$ standard deviation across runs. Additional details regarding datasets, architectures, and training procedures are provided in Appendix B. Our implementation and experimental code are publicly available at https://github.com/peiman-m/SConvCNP.

### 4.1 Synthetic 1-D Regression

We begin by evaluating models on four synthetic benchmarks generated from distinct stochastic processes: a GP with a Matérn–5/2 kernel, a GP with a periodic kernel, a sawtooth-wave generator, and a square-wave generator. For each benchmark, the parameters of the generative process—kernel hyperparameters for the GPs, frequency and direction for the sawtooth generator, and frequency and duty cycle for the square-wave generator—are sampled randomly. Complete experimental details are provided in Appendix B.1. Table 1 reports the average evaluation metrics over 1,000 test batches, each

Table 1: Comparison of predictive performance across methods on synthetically generated tasks. Lower RMSE and higher log-likelihood indicate better performance. For each metric and experimental setting, boldface denotes the top-two performing models.

| Metric | Data | Model | | | | | |
|---|---|---|---|---|---|---|---|
| | | CNP | AttCNP | TNP | TE-TNP | ConvCNP | SConvCNP |
| Log-likelihood ↑ | Matérn 5/2 | $-0.54_{\pm 0.01}$ | $-0.32_{\pm 0.00}$ | $\mathbf{-0.29}_{\pm 0.00}$ | $\mathbf{-0.28}_{\pm 0.00}$ | $-0.30_{\pm 0.01}$ | $\mathbf{-0.29}_{\pm 0.01}$ |
| | Periodic | $-1.20_{\pm 0.00}$ | $-0.87_{\pm 0.05}$ | $-0.76_{\pm 0.03}$ | $\mathbf{-0.57}_{\pm 0.02}$ | $-0.73_{\pm 0.01}$ | $\mathbf{-0.66}_{\pm 0.00}$ |
| | Sawtooth | $-0.90_{\pm 0.00}$ | $-0.90_{\pm 0.00}$ | $-0.90_{\pm 0.00}$ | $-0.90_{\pm 0.00}$ | $\mathbf{0.10}_{\pm 0.30}$ | $\mathbf{0.82}_{\pm 0.03}$ |
| | Square | $-1.39_{\pm 0.00}$ | $-1.41_{\pm 0.01}$ | $-1.33_{\pm 0.02}$ | $\mathbf{-1.17}_{\pm 0.06}$ | $-1.35_{\pm 0.04}$ | $\mathbf{-1.13}_{\pm 0.03}$ |
| RMSE ↓ | Matérn 5/2 | $0.50_{\pm 0.00}$ | $\mathbf{0.45}_{\pm 0.00}$ | $\mathbf{0.44}_{\pm 0.00}$ | $\mathbf{0.44}_{\pm 0.00}$ | $0.45_{\pm 0.00}$ | $0.45_{\pm 0.00}$ |
| | Periodic | $0.81_{\pm 0.00}$ | $0.65_{\pm 0.02}$ | $0.60_{\pm 0.02}$ | $\mathbf{0.50}_{\pm 0.01}$ | $0.57_{\pm 0.00}$ | $\mathbf{0.53}_{\pm 0.00}$ |
| | Sawtooth | $0.58_{\pm 0.00}$ | $0.58_{\pm 0.00}$ | $0.58_{\pm 0.00}$ | $0.58_{\pm 0.00}$ | $\mathbf{0.40}_{\pm 0.06}$ | $\mathbf{0.24}_{\pm 0.00}$ |
| | Square | $0.98_{\pm 0.00}$ | $1.00_{\pm 0.01}$ | $0.93_{\pm 0.02}$ | $\mathbf{0.81}_{\pm 0.01}$ | $0.89_{\pm 0.02}$ | $\mathbf{0.79}_{\pm 0.01}$ |

containing 16 tasks (each task defined in equation 2). For every task—independent of batch—both $x_c$ and $x_q$ are sampled independently from $\mathcal{U}[-3, 3]$. For each test batch, the number of context points, shared by all tasks within that batch, is drawn independently according to $n_c \sim \mathcal{U}[5, 25]$. The number of query points, however, is fixed at $n_q = 256$ for all the test tasks. As shown, SConvCNP consistently outperforms or closely matches strong baselines, including TE-TNP and ConvCNP, which represent the current state of the art. Figure 1 provides qualitative comparisons of predictive maps produced by SConvCNP, ConvCNP, and TE-TNP.

For the sawtooth-wave benchmark, we were unable to successfully train either TNP or TE-TNP. Across the configurations we attempted—including increased model capacity and multiple randomized initializations—their predictions consistently collapsed to zero. We hypothesize that this failure mode stems from *spectral bias*, the tendency of neural networks to favor low-frequency structure over high-frequency components [Rahaman et al., 2019, Ronen et al., 2019, Basri et al., 2020, Tancik et al., 2020, Fridovich-Keil et al., 2022]. Recent findings by Vasudeva et al. [2025] further suggest that this bias is *exacerbated* in transformer architectures compared with convolutional networks. This interpretation aligns with observations by Nguyen and Grover [2022], who originally introduced TNPs. They report degraded performance on GP samples drawn from a periodic kernel—reflected in poor log-likelihood scores—despite strong results on Matérn-kernel tasks. Interestingly, our experiments on periodic-kernel GP tasks (Table 1) do *not* replicate this limitation. A key distinction is that we employ the Efficient Query TNP (EQTNP) architecture of Feng et al. [2022], rather than the original TNP design [Nguyen and Grover, 2022] (see Appendix B.1.1 for architectural details). Finally, although sawtooth and square-wave functions share similar discontinuity and high-frequency characteristics, Table 1 shows that TNP and TE-TNP do *not* collapse when trained on square-wave signals. Investigating the source of this discrepancy falls beyond the scope of this work but represents an intriguing direction for future study.

## 4.2 Predator–Prey Dynamics

We next assess performance on simulated trajectories from a stochastic variant of the Lotka–Volterra predator–prey system [Lotka, 1910, Volterra, 1926], following the formulation in Bruinsma et al. [2023]. Let $U_t$ and $V_t$ denote the prey and predator populations at time $t$, respectively. Their dynamics evolve according to the stochastic Lotka–Volterra system

$$dU_t = \alpha U_t \, dt - \beta U_t V_t \, dt + \sigma U_t^\nu \, dB_t^{(1)}, \qquad dV_t = -\gamma U_t \, dt + \delta U_t V_t \, dt + \sigma V_t^\nu \, dB_t^{(2)},$$

where $B_t^{(1)}$ and $B_t^{(2)}$ are independent Brownian motions. In the deterministic component of the dynamics, $U_t$ grows exponentially at rate $\alpha$, while $V_t$ decays at rate $\gamma$. The bilinear interaction terms $\beta U_t V_t$ and $\delta U_t V_t$ model predation and the corresponding transfer of biomass from prey to predators. To account for stochastic fluctuations commonly observed in empirical population counts, the dynamics are augmented with multiplicative noise terms $\sigma U_t^\nu dB_t^{(1)}$ and $\sigma V_t^\nu dB_t^{(2)}$. Here, $\sigma$ controls the noise magnitude, while $\nu$ determines how the variability scales with population size.

Table 2: Comparison of predictive performance across methods on tasks constructed from the Lotka–Volterra system simulation. Lower RMSE and higher log-likelihood indicate better performance. For each metric and experimental setting, boldface denotes the top-two performing models.

| | CNP | AttCNP | TNP | TE-TNP | ConvCNP | SConvCNP |
|---|---|---|---|---|---|---|
| Log-likelihood ↑ | $-0.27_{\pm 0.01}$ | $0.08_{\pm 0.00}$ | $0.12_{\pm 0.00}$ | $\mathbf{0.16}_{\pm 0.00}$ | $\mathbf{0.14}_{\pm 0.00}$ | $\mathbf{0.14}_{\pm 0.00}$ |
| RMSE ↓ | $0.37_{\pm 0.00}$ | $\mathbf{0.31}_{\pm 0.00}$ | $\mathbf{0.31}_{\pm 0.00}$ | $\mathbf{0.30}_{\pm 0.0.00}$ | $\mathbf{0.31}_{\pm 0.00}$ | $\mathbf{0.30}_{\pm 0.00}$ |

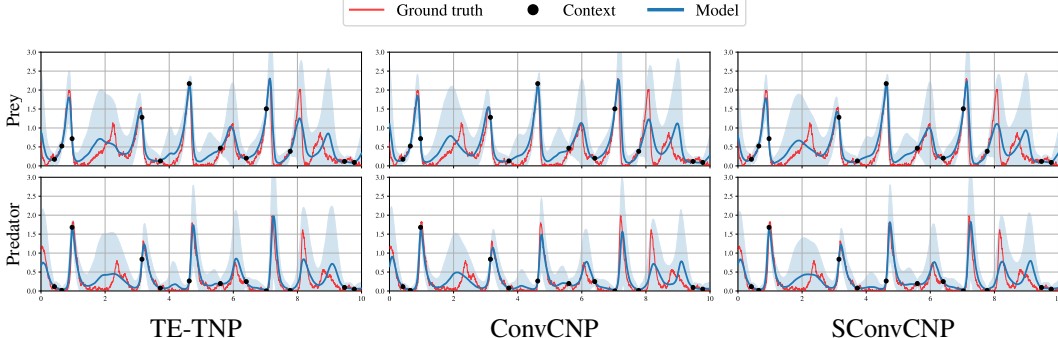

Figure 2: Illustrative predictions on simulated Lotka–Volterra predator–prey trajectories. Black markers denote the context observations. The blue curve shows the predictive mean, while the shaded region corresponds to the $\pm 2$ standard deviation interval of the Gaussian predictive distribution. Ground-truth population trajectories from the simulator are plotted in red.

We simulate trajectories over a dense uniform grid on $t \in [-10, 100)$, discarding the initial 10 years as burn-in. For each task, context points $x_c$ and query points $x_q$ are drawn independently and uniformly from the retained interval $[0, 100)$. Population values (corresponding to $y = [U_t, V_t]$) and time inputs (corresponding to $x = t$) are rescaled by factors of $0.01$ and $0.1$, respectively, before being fed to the models. A complete description of the experimental protocol is provided in Appendix B.2.

Table 2 reports average evaluation metrics over 1,000 test batches, each containing 16 tasks. For each batch, the number of context points—shared across all tasks—is sampled as $n_c \sim \mathcal{U}[5, 25]$, and the number of query points is fixed at $n_q = 256$. SConvCNP achieves log-likelihood performance comparable to ConvCNP, and both are competitive with TE-TNP. In terms of RMSE, SConvCNP matches TE-TNP, indicating that its predictive uncertainty intervals are slightly wider. Qualitative comparisons of predictive maps for the top three models appear in Figure 2.

### 4.3 Traffic Flow

For our third experiment, we evaluate on the California traffic-flow dataset from LargeST [Liu et al., 2023b]. This dataset comprises five years (2017–2021) of traffic measurements recorded every 5 minutes by approximately 8,600 loop-detector sensors deployed across California's highway network. We focus on the year 2020, where traffic patterns are expected to exhibit heightened variability due to the abrupt onset of the COVID-19 pandemic. For each sensor, we segment its year-long time series into non-overlapping 14-day windows and downsample each window by a factor of 6 (from 5-minute to 30-minute resolution). Each window is treated as a dense trajectory from which individual tasks are constructed. Additional details of the experimental setup are provided in Appendix B.3.

Table 3: Comparison of predictive performance across methods on tasks constructed from California traffic flow measurements. Lower RMSE and higher log-likelihood indicate better performance. For each metric and experimental setting, boldface denotes the top-two performing models.

| | CNP | AttCNP | TNP | TE-TNP | ConvCNP | SConvCNP |
|---|---|---|---|---|---|---|
| Log-likelihood ↑ | $1.73_{\pm 0.10}$ | $1.80_{\pm 0.01}$ | $1.93_{\pm 0.06}$ | $1.72_{\pm 0.02}$ | $\mathbf{1.98}_{\pm 0.02}$ | $\mathbf{2.05}_{\pm 0.02}$ |
| RMSE ↓ | $\mathbf{0.05}_{\pm 0.00}$ | $\mathbf{0.05}_{\pm 0.00}$ | $\mathbf{0.04}_{\pm 0.00}$ | $\mathbf{0.05}_{\pm 0.00}$ | $\mathbf{0.04}_{\pm 0.00}$ | $\mathbf{0.04}_{\pm 0.00}$ |

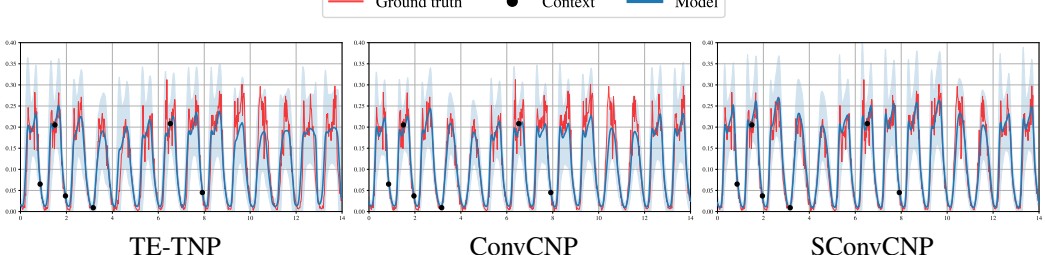

| TE-TNP | ConvCNP | SConvCNP |

Figure 3: Illustrative predictions on the California traffic-flow dataset. Black markers denote the context observations. The blue curve shows the predictive mean, while the shaded region corresponds to the $\pm 2$ standard deviation interval of the Gaussian predictive distribution. Ground-truth traffic measurements are shown in red.

Table 3 reports evaluation metrics averaged over 66,586 test tasks, constructed from 2,561 held-out test sensors and partitioned into batches of size 32. For each batch, the number of context points—shared across all tasks—is drawn as $n_c \sim \mathcal{U}[5, 25]$, and the number of query points is fixed at $n_q = 50$. Across all metrics, SConvCNP achieves the best performance, attaining the highest log-likelihood and lowest RMSE among the baselines. Figure 3 provides qualitative comparisons of the predictive maps produced by the three models.

## 4.4 Image Completion

For our final experiment, we evaluate model performance on an image-completion task formulated as a spatial regression problem, where the model maps 2D pixel coordinates to their corresponding intensity values. We use images from the Describable Textures Dataset (DTD; Cimpoi et al. [2014]), and construct each task from a processed $64 \times 64$ subsampled crop of an original image. Because each task contains a large number of context and query pixels, we were unable to fit TE-TNP within our computational budget—even after reducing its size—so we exclude it from this experiment. Additional experimental details are provided in Appendix B.4. Table 4 reports results averaged over 1,880 test tasks, evaluated in batches of 16. For each batch, the number of context points—shared across all tasks—is sampled as $n_c \sim \mathcal{U}[5, 1024]$; all remaining pixels serve as query points, so that $n_q = 4096 - n_c$. Among the baselines, SConvCNP achieves the highest log-likelihood and the second-lowest RMSE. Figure 4 presents qualitative comparisons of the predictive means produced by the top three models.

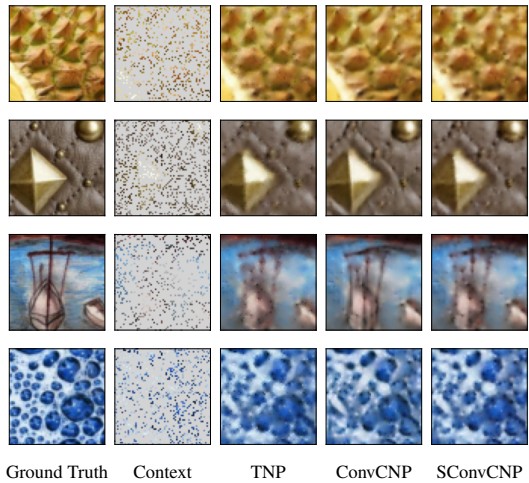

| Ground Truth | Context | TNP | ConvCNP | SConvCNP |

Figure 4: Illustrative model outputs for image completion on the DTD dataset. Gray pixels indicate query regions, while the remaining pixels serve as context observations. For each model, query regions are completed using the mean of the predictive distribution.

Table 4: Comparison of predictive performance across methods on image-completion tasks constructed from the DTD dataset. Lower RMSE and higher log-likelihood indicate better performance. For each metric and experimental setting, boldface denotes the top-two performing models..

| | CNP | AttCNP | TNP | ConvCNP | SConvCNP |
|---|---|---|---|---|---|
| Log-likelihood ↑ | $0.67_{\pm 0.01}$ | $1.39_{\pm .02}$ | $1.39_{\pm .05}$ | $\mathbf{1.48}_{\pm 0.00}$ | $\mathbf{1.50}_{\pm 0.02}$ |
| RMSE ↓ | $0.14_{\pm 0.00}$ | $\mathbf{0.08}_{\pm 0.00}$ | $\mathbf{0.06}_{\pm 0.00}$ | $\mathbf{0.08}_{\pm 0.00}$ | $\mathbf{0.08}_{\pm 0.00}$ |

# 5 Related Works

**Neural PDE Solvers** The high computational cost of traditional numerical PDE solvers has motivated the development of more efficient alternatives based on deep learning Gupta and Brandstetter [2022]. Among these, neural operators [Li et al., 2020b, Kovachki et al., 2023], and in particular Fourier neural operators (FNOs, Li et al., 2020a), have become popular. Numerous extensions have since been proposed: Helwig et al. [2023] introduced rotation- and reflection-equivariant FNOs; Gupta et al. [2021] developed multiwavelet neural operators by projecting kernels onto predefined polynomial bases; and Tran et al. [2021] reduced model complexity through separable Fourier representations. Xiao et al. [2024] proposed the Amortized FNO (AM-FNO), which substantially cuts parameter count by amortizing the Fourier kernel parameterization, while Qin et al. [2024] analyzed FNOs through the lens of spectral bias. Koshizuka et al. [2024] provided a mean-field theoretical perspective, and Zheng et al. [2024] introduced the Mamba Neural Operator, which couples global context via a state-space model to achieve linear complexity and representation equivalence. Bartolucci et al. [2023] examined continuous–discrete consistency, showing that it holds for convolutional neural operators [Raonic et al., 2023]. Additional works include orthogonal-attention eigenfunction methods for operator learning [Xiao et al., 2023] and hierarchical transformers with frequency-aware priors for resolution-invariant super-resolution [Luo et al., 2024].

**Function Space Inference** In non-parametric Bayesian modeling, GPs and deep GPs [Damianou and Lawrence, 2013] provide flexible function-space priors with well-calibrated uncertainty estimates, but their computational cost becomes prohibitive on large datasets. This challenge has motivated the development of Bayesian neural networks (BNNs; MacKay [1992], Hinton and Van Camp [1993], Neal [2012]), which combine neural network expressiveness with Bayesian uncertainty quantification. However, specifying meaningful priors over network weights remains notoriously difficult. Recent work therefore reframes Bayesian inference in neural networks as learning a posterior over the *functions* induced by stochastic weights [Wolpert, 1993, Qiu et al., 2023]. Variational implicit processes (VIPs; Ma et al. [2019], Santana et al. [2021], Ortega et al. [2022]) generalize GPs by defining implicit stochastic processes through latent variables, while functional variational BNNs (fBNNs; Sun et al. [2019]) enforce alignment between BNN-induced priors and target priors by minimizing a functional KL divergence—though this objective can be difficult to compute and, in some cases, ill-posed [Burt et al., 2020]. Subsequent work has aimed to address these limitations [Ma and Hernández-Lobato, 2021, Rudner et al., 2022, Wild et al., 2022, Rudner et al., 2023, Wu et al., 2024]. Orthogonally, energy-based models have been explored for representing stochastic processes in function space [Yang et al., 2020, Lim et al., 2022]. More recently, several approaches have extended diffusion and flow models to function spaces [Phillips et al., 2022, Dutordoir et al., 2023, Mathieu et al., 2023, Franzese et al., 2023, Lim et al., 2023, Kerrigan et al., 2024, Zhang and Scott, 2025].

# 6 Conclusion

In this work, we introduced spectral convolutional conditional neural processes (SConvCNPs), a class of conditional neural processes that incorporates ideas from operator learning—particularly Fourier neural operators—into the neural processes framework, with an emphasis on convolutional conditional neural processes. Empirical evaluations on a collection of synthetic and real-world datasets indicate that SConvCNPs perform comparably to strong baselines. These results suggest that integrating techniques from operator learning into neural processes is a viable direction for probabilistic function modeling and warrants further investigation.

# Acknowledgments

We thank Arman Hasanzadeh and Jonathan W. Siegel for their insightful feedback and constructive suggestions. We also acknowledge the Texas A&M High Performance Research Computing facility for providing the computational resources used in this study. Finally, we thank the anonymous reviewers for their valuable comments and suggestions, which helped improve the quality of this work.

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

# A  Discretization Sensitivity of the DFT

To illustrate the sensitivity of the discrete Fourier transform (DFT) to the underlying discretization—and how changes in grid geometry can degrade predictive performance—we consider a simple one-dimensional example. Let $0 < \Delta \ll 1$ denote a fixed discretization resolution, interpreted as the spacing between adjacent grid points. Define the grid

$$\mathcal{G}_1 = (x_0, \ldots, x_{m_1-1})$$

as a uniform discretization of the interval $[0, 1]$, where $m_1 = \lfloor 1/\Delta \rfloor$ and the points are ordered such that $x_n < x_{n+1}$ for all $n \in \{0, \ldots, m_1 - 2\}$. Let $h := \varphi_e[\mathcal{D}_c]$ denote the encoded latent function, and let its sampled values on $\mathcal{G}_1$ be

$$\big(h(x)\big)_{x \in \mathcal{G}_1} = \big(h(x_0), \ldots, h(x_{m_1-1})\big).$$

The DFT of $h$ on this grid is given by

$$\hat{\mathcal{F}}\Big\{\big(h(x)\big)_{x \in \mathcal{G}_1}\Big\}(k) = \sum_{n=0}^{m_1-1} h(x_n)\, e^{-i2\pi \frac{k}{m_1} n}, \qquad k \in \{0, \ldots, m_1 - 1\}.$$

This yields Fourier coefficients at the normalized frequencies

$$\Xi_1 = \left(\frac{k}{m_1}\right)_{k=0}^{m_1-1}.$$

As discussed in Section 2.1, the Fourier Neural Operator (FNO) implicitly ties its kernel parameterization to the specific frequency set $\Xi_1$ encountered during training. Now suppose the trained model is evaluated on a larger spatial domain, e.g. $[0, 2]$, while maintaining the same resolution $\Delta$. The corresponding grid

$$\mathcal{G}_2 = (x'_0, \ldots, x'_{m_2-1})$$

has size $m_2 = \lfloor 2/\Delta \rfloor$, and the associated DFT produces coefficients at normalized frequencies

$$\Xi_2 = \left(\frac{k}{m_2}\right)_{k=0}^{m_2-1}.$$

Since $m_2 \geq 2m_1$, we have $|\Xi_2| \geq 2|\Xi_1|$, and, crucially, the frequency locations themselves differ. Let $1 \leq k_{\max} \leq m_1$ denote the number of Fourier modes retained during training. Even when restricting attention to the lowest $k_{\max}$ modes, the frequencies $\{k/m_2\}_{k=0}^{k_{\max}-1}$ do not coincide with $\{k/m_1\}_{k=0}^{k_{\max}-1}$. This misalignment induces a spectral mismatch between training and evaluation, illustrating how changes in discretization—even at fixed resolution—can disrupt the learned Fourier parameterization and lead to degraded generalization.

# B  Experimental Details

All implementations are written in PyTorch [Paszke et al., 2019] and publicly available at https://github.com/peiman-m/SConvCNP. We used a single NVIDIA A100 GPU with 40 GB of memory for all the computations. Our code is based on the implementations of Ashman et al. [2024a] and Bruinsma et al. [2023].

## B.1  Synthetic 1-D Regression

### B.1.1  Model Architectures

This section details the architectures of all CNPs variants used in our experiments. Each model outputs a Gaussian predictive distribution parameterized by a mean and a pre-softplus [Dugas et al., 2000] scale. The pre-softplus value is passed through a softplus transformation, and a minimum noise term of $10^{-6}$ is then added to the resulting scale. The Gaussian factorizes across query points—as is standard for CNPs (see Section 2.2)—and, for multidimensional outputs, across output dimensions as well. Thus, predictions assume independence both across query locations and across components of each output vector. Unless otherwise noted, all nonlinearities use ReLU activations [Nair and Hinton, 2010].

**Conditional Neural Process (CNP):** The CNP encodes each context pair $(x_{c,k}, y_{c,k}) \in \mathcal{D}_c$ using separate input and output pathways. The input $x_{c,k}$ and output $y_{c,k}$ are first passed through distinct MLPs with two hidden layers each of dimension 256, producing 256-dimensional representations $\varepsilon_{c,k}^{(x)}$ and $\varepsilon_{c,k}^{(y)}$. Their concatenation $\varepsilon_{c,k} = [\varepsilon_{c,k}^{(x)}, \varepsilon_{c,k}^{(y)}]$ is then processed by a deeper MLP with six hidden layers of width 256 and a final 256-dimensional output layer, yielding an embedding for each context pair. Context embeddings $\varepsilon_{c,k}$ are averaged to form a single aggregated representation $\varepsilon_c$ of dimension 256. For prediction, each query input $x_{q,l}$ is concatenated with the context representation $\varepsilon_c$ and passed through a decoder MLP consisting of six hidden layers of dimension 256. The decoder's final output layer has dimensionality $2d_y$, parameterizing the mean and the log-scale parameters of the Gaussian predictive distribution.

**Attentive Conditional Neural Process (AttCNP).** The AttCNP implementation follows the deterministic architecture introduced by Kim et al. [2019]. The initial encoding stage mirrors that of the CNP: each context pair $(x_{c,k}, y_{c,k}) \in \mathcal{D}_c$ is processed by two separate MLPs (each with two hidden layers of width 256), producing embeddings $\varepsilon_{c,k}^{(x)}$ and $\varepsilon_{c,k}^{(y)}$. These are concatenated to form $\varepsilon_{c,k} = [\varepsilon_{c,k}^{(x)}, \varepsilon_{c,k}^{(y)}]$, which is subsequently passed through an additional two-layer MLP with hidden width 256. Departing from the CNP, the AttCNP applies self-attention to the set of context embeddings. We use two layers of multi-head self-attention, each with 8 heads (head dimension 32), a feedforward subnetwork with hidden width 256, residual connections, and pre-layer normalization for both the attention and feedforward blocks. The input and output dimensionality of each attention layer is fixed at 256. Query representations $\varepsilon_{q,l}$ are computed using a single layer of multi-head cross-attention between the (attended) context embeddings and the query embeddings, with the same configuration as above (8 heads of dimension 32, feedforward width 256, residual connections, and pre-normalization). The decoder is an MLP with six hidden layers of width 256. Its final layer outputs $2d_y$ units, corresponding to the mean and log-scale parameters of a Gaussian predictive distribution.

**Transformer Neural Process (TNP).** To apply the TNP, we first construct token representations for both context and query points. For each context pair $(x_{c,k}, y_{c,k}) \in \mathcal{D}_c$, we form the token

$$[x_{c,k}, \ y_{c,k}, \ 1],$$

where the final singleton "1" serves as a density flag indicating that the observation $y_{c,k}$ at $x_{c,k}$ is available. For each query location $x_{q,l}$, we instead construct

$$[x_{q,l}, \ \mathbf{0}, \ 0],$$

where the dummy zero-vector matches the shape of $y_{c,k}$ and the final "0" indicates the absence of an observation [Nguyen and Grover, 2022, Ashman et al., 2024a]. All tokens are then passed through a shared two-layer MLP (width 256), producing initial embeddings $\varepsilon_{c,k}$ and $\varepsilon_{q,l}$.

In the original TNP architecture [Nguyen and Grover, 2022], context and query embeddings are processed jointly by a transformer encoder. An attention mask prevents (i) interactions among queries and (ii) context→query attention, ensuring that context representations remain query-independent, while queries may attend to contexts. This design incurs quadratic complexity in the total sequence length,

$$\mathcal{O}((|\mathcal{D}_c| + |\mathcal{D}_q|)^2).$$

Because queries never attend to one another, Feng et al. [2022] introduced the Efficient Query TNP, which we adopt here. This variant first applies self-attention to context tokens to produce updated context embeddings, then processes each query using cross-attention over these updated contexts. The resulting two-branch structure (contexts processed twice, queries once) reduces the overall complexity to

$$\mathcal{O}\big(|\mathcal{D}_c|^2 + |\mathcal{D}_q|\,|\mathcal{D}_c|\big).$$

Our model uses six transformer layers with eight attention heads (head dimension 32), each followed by a feedforward subnetwork of width 256. We use the standard pre-norm architecture, applying layer normalization before both the attention and feedforward modules. All attention operations share a 256-dimensional embedding dimension. At each layer, we first apply context self-attention and then query–context cross-attention. Notably, the same multi-head attention parameters are used for both operations; the distinction arises only from which embeddings serve as queries, keys, and values (yielding self-attention when they coincide, and cross-attention otherwise). This is not a

modeling requirement—separate parameter sets could be used, as demonstrated in Appendix B.4.1. The final-layer query embeddings are passed to a decoder MLP with two hidden layers (width 256), whose output parameterizes a Gaussian predictive distribution via $2d_y$ units corresponding to the mean and log-scale.

**Translation-Equivariant Transformer Neural Process (TE-TNP).** The token construction in the TE-TNP closely follows that of the TNP, with a crucial modification to ensure translation equivariance: input locations are excluded from the token representations. Specifically, for each context pair $(x_{c,k}, y_{c,k}) \in \mathcal{D}_c$, we construct the token

$$[y_{c,k}, 1],$$

while for each query location $x_{q,l}$ we use the token

$$[\mathbf{0}, 0].$$

All tokens are passed through a shared two-layer MLP with width 256 to produce initial embeddings $\varepsilon_{c,k}$ and $\varepsilon_{q,l}$.

The TE-TNP replaces the standard multi-head attention mechanism with the translation-equivariant attention proposed by Ashman et al. [2024a]. For each attention head, we first compute the matrix of pairwise scaled dot products between token embeddings, analogous to conventional attention but independent of absolute input locations. In parallel, we compute the matrix of pairwise *differences* between token locations. We then concatenate the scaled dot-product scores from all heads with the corresponding pairwise location differences. This augmented pairwise similarity representation is processed by an MLP—acting as an implicit kernel—with two hidden layers of width 256 and an output dimension equal to the number of attention heads, yielding the final translation-equivariant attention logits.

Apart from this modified attention module, the remainder of the architecture and computational pipeline follows that of the TNP.

**Convolutional Conditional Neural Process (ConvCNP):** For the ConvCNP, we begin by determining, for each input dimension, the minimum and maximum coordinates observed across both the context and query sets. These extrema are expanded by a small margin of 0.1 in each dimension. The interval between the expanded minima and maxima is then uniformly discretized at a resolution of 64 points per unit. When necessary—for example, to satisfy the CNN's minimum grid-size requirements—the discretization range is further enlarged while maintaining the same resolution. The resulting one-dimensional grids are combined via a Cartesian product to yield a uniform grid $\mathcal{G}$.

The functional embedding in equation 3 is evaluated on this grid. The Gaussian kernels used in this embedding are initialized with per-dimension length scales set to twice the grid resolution, i.e., $2/64$. For multi-dimensional inputs and outputs, the model uses separate length scales for each dimension. Following Bruinsma et al. [2023], the embedding is further normalized using a *density channel*, defined as

$$\text{Density}(x) = \sum_{(x_c, y_c) \in \mathcal{D}_c} \psi_e(x - x_c).$$

The functional and density channels are concatenated to form the final grid representation

$$\left( \text{Density}(x_g), \frac{\sum_{(x_c, y_c) \in \mathcal{D}_c} \phi(y_c) \, \psi_e(x_g - x_c)}{\text{Density}(x_g)} \right)_{x_g \in \mathcal{G}}. \tag{5}$$

Each grid point is processed independently by an MLP with two hidden layers of width 128. The resulting features are passed to a CNN based on a U-Net architecture [Ronneberger et al., 2015], consisting of six residual convolutional blocks (kernel size 11, stride 2, 128 channels) in the encoder, followed by a symmetric sequence of six transposed convolutional blocks in the decoder, with skip connections following the design of Bruinsma et al. [2023]. Since the U-Net downsamples the spatial resolution by a factor of 64, we ensure that the constructed grid size is divisible by 64. For the one-dimensional benchmarks considered in this work, this is achieved by symmetrically enlarging the discretization interval when necessary so that the CNN output aligns exactly with the input grid.

To obtain predictive distribution parameters at query locations—which may not lie on the grid $\mathcal{G}$—an interpolation scheme analogous to equation 5 is applied, except that the weighted sum over the

feature-map values is not normalized. For this interpolation, we employ a separate Gaussian kernel with learnable length-scale parameters, distinct from the kernel used in the functional embedding. The resulting query-specific embeddings are then processed by a decoder MLP with two hidden layers (width 128), whose output consists of $2d_y$ units parameterizing the mean and log-scale of a Gaussian predictive distribution.

**Spectral Convolutional Conditional Neural Process (SConvCNP)**  The SConvCNP replaces the ConvCNP's U-Net backbone with a U-shaped Fourier Neural Operator (FNO) architecture [Li et al., 2020a, Rahman et al., 2022a]. A residual Fourier block is denoted

$$\mathrm{F}(c_{\text{in}}, c_{\text{out}}, s_{\text{in}}, s_{\text{out}}, m_{\text{f}}),$$

where $c_{\text{in}}$ and $c_{\text{out}}$ are channel dimensions, $s_{\text{in}}$ and $s_{\text{out}}$ are spatial sizes, and $m_{\text{f}}$ is the number of retained Fourier modes.

The representation from equation 5, augmented with positional encodings (grid coordinates), is first processed by an MLP with one hidden layer of width 64 and GELU activations [Hendrycks and Gimpel, 2016]. Its output is then passed through the following sequence of residual Fourier blocks:

- $\mathrm{L}_1 = \mathrm{F}(64, 128, |\mathcal{G}|, \lfloor|\mathcal{G}|/2\rfloor, 32)$
- $\mathrm{L}_2 = \mathrm{F}(128, 128, \lfloor|\mathcal{G}|/2\rfloor, \lfloor|\mathcal{G}|/4\rfloor, 32)$
- $\mathrm{L}_3 = \mathrm{F}(128, 256, \lfloor|\mathcal{G}|/4\rfloor, 32)$
- $\mathrm{L}_4 = \mathrm{F}(256, 128, \lfloor|\mathcal{G}|/2\rfloor, 32)$
- $\mathrm{L}_5 = \mathrm{F}(256, 128, |\mathcal{G}|, 32)$

Layers $\mathrm{L}_1 \rightarrow \mathrm{L}_2 \rightarrow \mathrm{L}_3 \rightarrow \mathrm{L}_4$ form the contractive–expansive path. The final block $\mathrm{L}_5$ receives the channel-wise concatenation of the outputs of $\mathrm{L}_4$ and $\mathrm{L}_1$, yielding the U-shaped skip connection. Its output is concatenated with the initial MLP features and processed by a final MLP (one hidden layer, width 128, GELU) to produce the SConvCNP representation for the decoder.

**Parameter Count.**  Table 5 summarizes the number of learnable parameters for all models used in our 1D synthetic regression experiments.

Table 5: Learnable parameter counts for all models used in the 1D synthetic regression experiments.

|  | CNP | AttCNP | TNP | TE-TNP | ConvCNP | SConvCNP |
|---|---|---|---|---|---|---|
| Number of parameters (million) | 1.3 | 2.1 | 2.6 | 3.1 | 3.8 | 3.7 |

**Forward run time.**  Table 6 reports the forward-pass runtime (in seconds) for a batch of 16 tasks during both training and validation across all models evaluated in our 1D synthetic regression experiments. As described in Appendix B.1.2, the number of query points is fixed to 256 during validation, whereas during training it is sampled uniformly from $\mathcal{U}[5, 25]$.

Table 6: Average forward-pass runtime (in seconds) for a batch of 16 tasks during training and validation across all models in the 1D synthetic regression setting.

|  | CNP | AttCNP | TNP | TE-TNP | ConvCNP | SConvCNP |
|---|---|---|---|---|---|---|
| Train | 0.003 | 0.004 | 0.007 | 0.014 | 0.009 | 0.009 |
| Validation | 0.006 | 0.009 | 0.013 | 0.023 | 0.033 | 0.030 |

### B.1.2  Data and Experimental Setup

We evaluate our methods on four families of synthetic 1D regression tasks, each defined by a distinct stochastic generative process: a Gaussian process (GP) with a Matérn–5/2 kernel, a GP with a periodic kernel, a sawtooth-wave generator, and a square-wave generator. GP tasks are sampled using GPyTorch [Gardner et al., 2018]. All processes include independent Gaussian observation noise.

**Generative Processes.** For each task family, process-specific hyperparameters are sampled independently at the task level.

- **GP with Matérn–5/2 kernel.** Functions are sampled from $f \sim \mathcal{GP}(0, k_{\mathrm{m5/2}} + \sigma_0^2 I)$, where

$$k_{\mathrm{m5/2}}(x, x') = \frac{2^{-1.5}}{\Gamma(2.5)} \left( \frac{\sqrt{5}}{\lambda} |x - x'| \right)^{2.5} K_{2.5} \left( \frac{\sqrt{5}}{\lambda} |x - x'| \right)$$

  and $K_{2.5}$ is the modified Bessel function of the second kind. The lengthscale is sampled as $\lambda \sim \mathcal{U}[0.25, 1)$. Observation noise standard deviation is $\sigma_0 = 0.1$.

- **GP with periodic kernel.** Functions are sampled from $f \sim \mathcal{GP}(0, k_{\mathrm{p}} + \sigma_0^2 I)$, where

$$k_{\mathrm{p}}(x, x') = \exp\left( -\frac{2 \sin^2(\pi |x - x'|/\rho)}{\lambda^2} \right)$$

  with period $\rho \sim \mathcal{U}[0.5, 2)$ and lengthscale $\lambda \sim \mathcal{U}[0.25, 1)$. Observation noise standard deviation is $\sigma_0 = 0.1$.

- **Sawtooth wave.** Functions are sampled from $f \sim \mathcal{GP}(m_{\mathrm{saw}}, \sigma_0^2 I)$, where the mean function is

$$m_{\mathrm{saw}}(x) = 2\left( (\xi\,(ux - c)) \bmod 1 \right) - 1$$

  with frequency $\xi \sim \mathcal{U}[0.5, 5)$, direction $u \in \{+1, -1\}$ (sampled uniformly), and phase offset $c \sim \mathcal{U}[0, 1)$. Observation noise standard deviation is $\sigma_0 = 0.05$.

- **Square wave.** Functions are sampled from $f \sim \mathcal{GP}(m_{\mathrm{sq}}, \sigma_0^2 I)$, where the mean function is

$$m_{\mathrm{sq}}(x) = 2 \, \mathbb{1}_{\{((\xi x - c) \bmod 1) < D\}} - 1$$

  with frequency $\xi \sim \mathcal{U}[0.5, 5)$, duty cycle $D \sim \mathcal{U}[0.25, 0.75)$, and phase offset $c \sim \mathcal{U}[0, 1)$. Observation noise standard deviation is $\sigma_0 = 0.05$.

Each model is trained for 500 epochs using AdamW [Loshchilov and Hutter, 2017] with a learning rate of $5 \times 10^{-4}$. We apply gradient clipping with a maximum norm of 0.5 [Pascanu et al., 2013]. Each epoch consists of 1000 iterations, and every iteration processes a batch of 16 tasks, yielding a total of 8 million on-the-fly sampled training tasks. For each batch, the numbers of context and query points are independently drawn as $n_c \sim \mathcal{U}[5, 25)$ and $n_q \sim \mathcal{U}[5, 25)$. These values are shared across all tasks in the batch. Input locations are sampled uniformly and independently from $[-3, 3)$ for each task.

For validation, we use a fixed set of 4,096 tasks, organized into 256 batches of 16 tasks. In these tasks, the number of query points is fixed at 256, while the number of context points is sampled using the same procedure as during training. Testing follows the same configuration as validation, except that we evaluate on 16,000 test tasks. Unlike the dynamically generated training tasks, the validation and test sets remain fixed across all experiments and runs.

## B.2 Predator-Prey Model

### B.2.1 Model Architectures

The model architectures and parameter counts closely follow those described in B.1.1. The only modification concerns the discretization of the functional embedding in the ConvCNP and the SConvCNP. Specifically, we first expand the range defined by the minima and maxima of all context and query coordinates by a margin of 0.5 in each dimension. The resulting interval is then uniformly discretized at a resolution of 48 points per unit length.

### B.2.2 Data and Experimental Setup

We generate data from a stochastic variant of the Lotka–Volterra predator–prey system [Lotka, 1910, Volterra, 1926], following the formulation of Bruinsma et al. [2023]. Let $U_t$ and $V_t$ denote the prey and predator populations at time $t$. Their dynamics evolve according to

$$dU_t = \alpha U_t \, dt - \beta U_t V_t \, dt + \sigma U_t^{\nu} \, dB_t^{(1)},$$

$$dV_t = -\gamma U_t \, dt + \delta U_t V_t \, dt + \sigma V_t^{\nu} \, dB_t^{(2)},$$

where $B_t^{(1)}$ and $B_t^{(2)}$ are independent Brownian motions. The deterministic drift recovers classical predator–prey behavior: prey grow exponentially at rate $\alpha$ in the absence of predators, while predators decline at rate $\gamma$ without prey. The interaction terms $\beta U_t V_t$ and $\delta U_t V_t$ model predation and predator reproduction. Stochasticity enters through the multiplicative noise terms $\sigma U_t^\nu$ and $\sigma V_t^\nu$, where $\sigma$ controls noise intensity and $\nu$ determines how fluctuations scale with population size. Setting $\nu = 1$ yields noise proportional to population level, while $\nu < 1$ and $\nu > 1$ induce sublinear and superlinear scaling, respectively.

Table 7: Parameter distributions for the stochastic Lotka–Volterra equations.

| Parameter | Distribution |
|---|---|
| Initial condition $U_{-10}$ | $\mathcal{U}([5, 100])$ |
| Initial condition $V_{-10}$ | $\mathcal{U}([5, 100])$ |
| $\alpha$ | $\mathcal{U}([1.0, 5.0])$ |
| $\beta$ | $\mathcal{U}([0.04, 0.08])$ |
| $\gamma$ | $\mathcal{U}([1.0, 2.0])$ |
| $\delta$ | $\mathcal{U}([0.04, 0.08])$ |
| $\sigma$ | $\mathcal{U}([0.5, 10])$ |
| $\eta$ | $\mathcal{U}([1, 5])$ |
| $\nu$ | Fixed at $1/6$ |

We simulate trajectories using the parameter distributions in Table 7 over a total of 110 years, discarding the first 10 years as burn-in to reduce sensitivity to initial conditions. Integration is performed using the Euler–Maruyama method with time step $\Delta t = 0.022$, producing 5000 solver steps over the interval $t \in [-10, 100]$. Although the solver runs at this finer resolution, we record states only on a uniform grid with spacing $0.05$, yielding approximately 2200 selected time points spanning $[-10, 100)$.

To construct a task from a simulated trajectory, input locations $t$ are sampled uniformly and independently from the recorded time points in $[0, 100)$ and paired with the corresponding values $(U_t, V_t)$. We rescale time by a factor of $0.1$, mapping the 100-year interval to $[0, 10]$, and rescale population sizes by multiplying the values by $0.01$.

Models are trained for 500 epochs using AdamW with a learning rate of $10^{-4}$. We apply gradient clipping with a maximum norm of $0.5$. Each epoch consists of 1,000 iterations, each processing a batch of 16 tasks, yielding a total of 8 million on-the-fly sampled training tasks. For every batch, the numbers of context and query points are drawn independently as $n_c \sim \mathcal{U}[5, 25]$ and $n_q \sim \mathcal{U}[5, 25]$; the sampled values are shared across all tasks within the batch.

For validation, we use a fixed set of 4,096 tasks, arranged into 256 batches of 16 tasks. In these tasks, the number of query points is fixed at 256, while the number of context points is sampled using the same procedure as in training. Testing follows the same protocol as validation, except that we evaluate on 16,000 test tasks. Unlike the dynamically generated training tasks, the randomization used to construct the validation and test tasks is fixed across all experiments and runs.

## B.3 Traffic Flow

### B.3.1 Model Architectures

The model architectures follow those described in Section B.1.1. For the discretization of the functional embedding in both the ConvCNP and the SConvCNP, we use an expanded margin of $0.5$ and a resolution of 48 grid points per unit length.

### B.3.2 Data and Experimental Setup

We use the California traffic-flow dataset from LARGEST [Liu et al., 2023b], a large-scale benchmark for traffic forecasting, available at https://www.kaggle.com/datasets/liuxu77/largest. The dataset contains traffic-flow measurements from 8,600 loop-detector sensors deployed across

California's highway network, collected at 5-minute intervals between 2017 and 2021. In our experiments, we restrict attention to data from the year 2020.

As a preprocessing step, we discard sensors with more than 50% missing values. For the remaining sensors, missing entries are filled via linear interpolation between observed measurements, with leading and trailing gaps completed using forward and backward propagation, respectively. Sensors are then randomly partitioned into training, validation, and test sets using a 6:1:3 split.

Each sensor's time series is segmented into non-overlapping, continuous 14-day windows (4,032 time steps), yielding 26 windows per sensor. Within each window, timestamps are reset to start at 0 and increase in 5-minute increments. We subsequently downsample each window by a factor of 6, resulting in a temporal resolution of 30 minutes. Time indices are rescaled to units of days by dividing by $(60 \times 24)$. Finally, traffic-flow values are normalized to the $[0, 1]$ range using min–max statistics computed from the training set.

The processed dataset contains:

- **Training:** 5,119 sensors $\rightarrow$ 133,094 windows
- **Validation:** 853 sensors $\rightarrow$ 22,178 windows
- **Test:** 2,561 sensors $\rightarrow$ 66,586 windows

Models are trained for 100 epochs using AdamW with a learning rate of $10^{-4}$. We apply gradient clipping with a maximum norm of $0.5$. Each epoch consists of approximately 4,160 iterations, each processing a batch of 32 tasks. For every batch, the numbers of context and query points are drawn independently as $n_c \sim \mathcal{U}[5, 25]$ and $n_q \sim \mathcal{U}[5, 25]$; the sampled values are shared across all tasks within the batch. A task is formed by independently and uniformly sampling $n_c + n_q$ time steps from a window without replacement. Training tasks are sampled on-the-fly from training windows. For validation and testing, we pre-generate fixed sets of tasks:

- **Validation:** 22,178 tasks (batched in 32), with $n_q = 256$ and $n_c$ sampled as in training.
- **Test:** 66,586 tasks (batched in 32), with $n_q = 50$ and $n_c$ sampled as in training.

## B.4    Image Completion

### B.4.1    Model Architectures

All models output Gaussian predictive distributions parameterized by a mean and a pre-softplus standard deviation. Because pixel values are normalized to $[0, 1]$, the predicted mean is passed through a sigmoid. The predicted standard deviation is obtained by applying a softplus to the network output, scaling the result by $0.99$, and adding a minimum noise of $0.01$.

**Conditional Neural Process (CNP).**    The CNP architecture follows the design used in previous experiments (see Section B.1.1), with the only modification being an increase in the MLP hidden size from $256$ to $512$.

**Attentive Conditional Neural Process (AttCNP).**    The AttCNP architecture is likewise based on the design used in earlier experiments (see Section B.1.1). The MLP hidden size is increased from $256$ to $512$. For the self-attention applied to the set of context embeddings, we increase the per-head dimension from $32$ to $64$, set the input/output dimension of each attention layer to $512$, and increase the feedforward subnetwork width from $256$ to $512$. The same modifications apply to the multi-head cross-attention used to update query embeddings (eight heads of dimension $64$ and feedforward width $512$).

**Transformer Neural Process (TNP).**    As with the other models, all MLP hidden sizes are increased from $256$ to $512$. We adopt the Efficient Query TNP design described in Section B.1.1, which employs a two-branch structure in which context embeddings are processed twice and query embeddings once. In contrast to previous sections, we allocate *separate* parameters for each branch: at each layer, the multi-head self-attention over contexts and the multi-head cross-attention between queries and contexts no longer share parameters. For each multi-head attention module, the feedforward width is increased from $256$ to $512$, and all input/output embedding dimensions are set to $512$. All other configurations follow Section B.1.1.

**Convolutional Conditional Neural Process (ConvCNP).** Because the data lie on a regular grid, we use the on-the-grid implementation of the ConvCNP [Gordon et al., 2019], eliminating the discretization step and improving computational efficiency. Let I denote an incomplete image, with unobserved pixels filled with dummy values, and let $M_c$ be the binary mask indicating observed (context) pixels. For multi-channel images, the mask is broadcast along the channel dimension.

As in Section B.1.1, the output of the convolutional deep set module has two components: (i) a density channel capturing the spatial distribution of context pixels, and (ii) a kernel-smoothed representation. The kernel is implemented via a 2D convolutional layer (kernel size 11, $d_y$ input channels, 128 output channels, no bias) with a nonnegativity constraint enforced by taking absolute values of the learned weights during the forward pass [Gordon et al., 2019].

The density channel is obtained by convolving this modified filter with the mask $M_c$. The kernel-smoothed component is computed by first multiplying I elementwise with $M_c$ (setting non-context pixels to zero) and then applying the same modified convolution. Following Gordon et al. [2019], we omit normalization by the density channel, as it did not provide empirical benefits and occasionally introduced instability; however, we retain the positivity constraint on the kernel.

The resulting representation is processed pointwise by an MLP with two hidden layers of width 128. These features are then fed into a ResNet-style CNN [He et al., 2016] consisting of six residual convolutional blocks (kernel size 11, 128 channels), following the implementation of Bruinsma et al. [2023]. The embeddings corresponding to query pixels are finally passed to a decoder MLP with two hidden layers of width 128.

**Spectral Convolutional Conditional Neural Process (SConvCNP).** The SConvCNP mirrors the on-the-grid ConvCNP architecture described above, with the distinction that the ResNet backbone is replaced by a U-shaped FNO. To construct positional information, we uniformly discretize the interval $[-1, 1]$ along each spatial axis into grids whose sizes match the corresponding image dimensions. Their Cartesian product yields the 2D positional encoding, which is concatenated with the output of the convolutional deep set. This concatenated tensor is then passed pointwise through an MLP with one hidden layer of width 128 and GELU activations.

Let $s_h$ and $s_w$ denote the image height and width, respectively. Using the notation introduced in Section B.1.1, the operator consists of the following sequence of residual Fourier blocks:

- $L_1 = F(512, 512, (s_h, s_w), (s_h, s_w), 32)$
- $L_2 = F(512, 512, (s_h, s_w), (\lfloor s_h/4 \rfloor, \lfloor s_w/4 \rfloor), 8)$
- $L_3 = F(512, 512, (\lfloor s_h/4 \rfloor, \lfloor s_w/4 \rfloor), (\lfloor s_h/16 \rfloor, \lfloor s_w/16 \rfloor), 2)$
- $L_4 = F(512, 512, (\lfloor s_h/16 \rfloor, \lfloor s_w/16 \rfloor), (\lfloor s_h/16 \rfloor, \lfloor s_w/16 \rfloor), 2)$
- $L_5 = F(512, 512, (\lfloor s_h/16 \rfloor, \lfloor s_w/16 \rfloor), (\lfloor s_h/4 \rfloor, \lfloor s_w/4 \rfloor), 8)$
- $L_6 = F(512, 512, (\lfloor s_h/4 \rfloor, \lfloor s_w/4 \rfloor), (s_h, s_w), 16)$
- $L_7 = F(512, 512, (s_h, s_w), (s_h, s_w), 32)$

The forward pass follows a standard U-shaped pattern. Layers $L_1 \rightarrow L_2 \rightarrow L_3 \rightarrow L_4$ are applied sequentially. On the upward path, $L_5$ receives the channel-wise concatenation of the outputs of $L_4$ and $L_3$. Similarly, $L_6$ receives the concatenation of the outputs of $L_5$ and $L_2$, and $L_7$ receives the concatenation of the outputs of $L_6$ and $L_1$. The output of $L_7$ is then concatenated with the features from the initial MLP and fed into a final MLP (one hidden layer, width 128, GELU) to yield the SConvCNP representation consumed by the decoder.

**Parameter Count.** Table 8 summarizes the number of learnable parameters for all models evaluated in our image completion experiments.

Table 8: Learnable parameter counts for all models evaluated in the image completion experiments.

|  | CNP | AttCNP | TNP | ConvCNP | SConvCNP |
|---|---|---|---|---|---|
| Number of parameters (million) | 5 | 8.4 | 13.7 | 12.2 | 14.2 |

### B.4.2 Data and Experimental Setup

For our image-completion experiments, we use the Describable Textures Dataset (DTD; Cimpoi et al. [2014]). We adopt the standard DTD split, which contains 1,880 images each for training, validation, and testing. All images are RGB, with heights ranging from 231 to 778 pixels and widths from 271 to 900 pixels. During training, validation, and testing, we sample from each image a random $192 \times 192$ crop, which we then downsample to $64 \times 64$. Pixel coordinates of the resulting $64 \times 64$ grid along each axis are linearly mapped to $[-1, 1]$, and pixel intensities are normalized to $[0, 1]$ independently across channels.

Models are trained for 500 epochs using AdamW with a learning rate of $10^{-4}$. We apply gradient clipping with a maximum norm of $0.5$. At each epoch, the model processes batches of 16 tasks. For each batch, we draw the number of context pixels as $n_c \sim \mathcal{U}[5, 1024]$, and treat all remaining pixels as queries, i.e. $n_q = 64 \times 64 - n_c$. A single sampled value of $n_c$ is shared across all tasks within the batch.

### B.5 Ablation Studies

We conduct a series of ablation studies to assess the contribution of three core design choices in the SConvCNP:

1. the number of retained Fourier modes,
2. the discretization resolution of the functional embedding, and
3. the use of positional encodings.

Experiments are conducted on two families of one-dimensional functions: samples drawn from a GP with a Matérn–$5/2$ kernel, and a deterministic sawtooth waveform. Training follows the protocol described in Section B.1, with the only difference being the use of a smaller SConvCNP model (3.1M parameters, 32 Fourier modes). Results are reported over 8,096 test tasks, and the main findings are summarized below.

Table 9: Predictive log-likelihood of SConvCNP for different numbers of Fourier modes $m$.

|  | $m = 8$ | $m = 16$ | $m = 32$ |
|---|---|---|---|
| Matérn 5/2 | $-0.29_{\pm 0.00}$ | $-0.29_{\pm 0.00}$ | $-0.29_{\pm 0.00}$ |
| Sawtooth wave | $-0.14_{\pm 0.03}$ | $0.20_{\pm 0.02}$ | $0.80_{\pm 0.03}$ |

**Number of Fourier modes.** Increasing the number of retained Fourier modes yields substantial improvements for the sawtooth signal, while having a negligible effect on Matérn–$5/2$ functions (Table 9). This behavior aligns with the spectral properties of the underlying functions. The sawtooth wave exhibits Fourier amplitudes that decay as $1/\xi$, leaving significant energy at high frequencies and necessitating a large number of modes for accurate reconstruction. In contrast, Matérn–$5/2$ samples have power spectra that decay as $1/\xi^6$, resulting in extremely weak high-frequency content; consequently, a small set of low-frequency modes suffices to capture nearly all signal energy.

Table 10: Predictive log-likelihood of SConvCNP for different discretization resolutions (number of points per unit).

|  | 16 | 32 | 64 |
|---|---|---|---|
| Matérn 5/2 | $-0.30_{\pm 0.00}$ | $-0.29_{\pm 0.00}$ | $-0.29_{\pm 0.00}$ |
| Sawtooth wave | $0.03_{\pm 0.02}$ | $0.26_{\pm 0.08}$ | $0.20_{\pm 0.02}$ |

**Discretization resolution.** A finer discretization improves performance for the sawtooth signal but has little impact on Matérn–$5/2$ samples (Table 10). The slowly decaying spectrum of the sawtooth implies a high effective Nyquist frequency, requiring dense sampling to resolve sharp transitions. By

contrast, the spectral mass of Matérn–5/2 functions is concentrated at lower frequencies, so coarser discretizations can adequately capture the relevant structure.

Table 11: Predictive log-likelihood of SConvCNP with and without positional encoding.

|  | with positional encoding | without positional encoding |
| --- | --- | --- |
| Matérn 5/2 | $-0.29_{\pm 0.00}$ | $-0.31_{\pm 0.00}$ |
| Sawtooth wave | $0.80_{\pm 0.03}$ | $0.67_{\pm 0.02}$ |

**Positional encoding.** Incorporating positional encodings consistently improves predictive performance, with particularly pronounced gains on the sawtooth tasks (Table 11). This suggests that explicit location information helps disambiguate high-frequency and non-smooth patterns that are not fully captured by fully translation equivariant features alone in the SConvCNP.

