# OpenReview forum: "Spectral Convolutional Conditional Neural Processes"
_NeurIPS.cc/2025/Conference — NeurIPS 2025 poster_

### Official Review · Reviewer_c7aT · 2025-06-22

**Clarity:** 3
**Significance:** 2
**Originality:** 3
**Rating:** 5
**Confidence:** 4

**Summary:**

The paper introduces Spectral Convolutional Conditional Neural Processes. It is a novel extension of Convolutional Conditional Neural Processes (ConvCNPs) that leverages spectral methods to address limitations in capturing long-range dependencies in irregularly sampled data. Note tha Traditional ConvCNPs use local discrete kernels, which struggles with global patterns due to finite receptive fields. SConvCNPs replace these with global convolutional filters parameterized in the frequency domain, inspired by Fourier Neural Operators (FNOs). This approach efficiently models stationary stochastic processes by focusing on low-frequency components, reducing computational costs while improving performance.
In terms of evaluation, The authors validates SConvCNPs on synthetic 1D regression tasks and real-world benchmarks, e.g. Lotka-Volterra dynamics, traffic flow data. Results show superior performance over baselines in log-likelihood and RMSE, particularly for periodic functions.

**Questions:**

See Weakness part.

**Ethical Concerns:**

["NO or VERY MINOR ethics concerns only"]

**Final Justification:**

After reading the rebuttal, my concerns are addressed. The work contributes a novel spectral convolution for neural processes, which is meaningful in the field.

**Limitations:**

See Weakness part.

**Paper Formatting Concerns:**

No concerns.

**Quality:**

3

**Strengths And Weaknesses:**

**Strengths:**

Compared with ConvCNPs, the prorponsed SConvCNP benefits from (1) It considers the Global dependency modeling. It addresses ConvCNPs' limitation of local receptive fields by leveraging spectral convolutions. (2) It presents outstanding empirical results: SConvCNHP outperforms baselines on synthetic and real-world tasks, especially for periodic patterns. Spectral truncation reduces computational overhead compared to large spatial kernels. (3) There constitutes theoretical insights. Builds on well-established frameworks (FNOs, CNPs) with clear motivation.

**Weakness:**

(1) I think FFT-based methods depend on grid resolution, which could affect robustness in highly irregular data. This part can be better analyzed. (2) For nonstationary cases, I am wondering SConvCNP’s performance over existing methods. If not, it would be better to include the limitation section. (3) Scalability issue in optimization objective. This work narrows the scope into ConvCNP. In terms of optimization objective, can the mentioned Fourier neural operator be easily integrated into vanilla NPs, e.g., approximate ELBO [1], importance weighted NP methods [2-3] or mixture-of-expert methods [4]? This can be discussed. (4) Some typos, such as missing reference number in Line 216 LargeST [?]

**References:**

[1]Garnelo, Marta, et al. "Neural processes." arXiv preprint arXiv:1807.01622 (2018).

[2]Wang, Qi, Marco Federici, and Herke van Hoof. "Bridge the inference gaps of neural processes via expectation maximization." arXiv preprint arXiv:2501.03264 (2025).

[3]Foong, Andrew, et al. "Meta-learning stationary stochastic process prediction with convolutional neural processes." Advances in Neural Information Processing Systems 33 (2020): 8284-8295.

[4]Wang, Qi, and Herke Van Hoof. "Learning expressive meta-representations with mixture of expert neural processes." Advances in neural information processing systems 35 (2022): 26242-26255.

---

> ### Author Rebuttal · Authors · 2025-07-31
>
> We thank the reviewer for their positive assessment of our work and for highlighting the strengths of our approach—including the motivation for global dependency modeling, promising empirical performance on periodic tasks, and the theoretical grounding in established operator learning frameworks.
>
> **(1) Robustness to Irregular Sampling and Grid Resolution**
>
> We appreciate the reviewer’s insightful comment. It is true that FFT-based methods typically rely on regular grids, which may raise concerns in the context of irregularly sampled data. To address this, we adopt the approach used in ConvCNPs, wherein context observations are first projected onto a fixed-resolution uniform grid via a learned interpolation kernel. Spectral convolutions are then applied to this gridded representation. This step allows our model to benefit from global Fourier structure while retaining robustness to the irregularity of inputs.
>
> **(2) Scalability and Integration with Other NP Objectives**
>
> We agree that extending SConvCNP to the class of latent NP families is a promising direction. While our current focus is on deterministic ConvCNP-style architectures, the proposed spectral module is compatible with latent-variable extensions. These extensions would enhance uncertainty modeling and enable richer predictive distributions.
>
> **(3) Errors and Typos**
>
> We thank the reviewer for noting the typo on line 216 (“LargeST [?]”). This will be corrected in the revision. We will also perform a comprehensive proofreading pass to address any remaining typographical issues and ensure clarity and consistency throughout the manuscript.

---

> > ### Comment · Reviewer_c7aT · 2025-08-01
> >
> > I thank the author for the detailed responses. After reading the rebuttal, I think my concerns are addressed and increase the score to accept.

---

> > > ### Author Response · Authors · 2025-08-02
> > >
> > > Thank you for your thoughtful feedback and for reconsidering your score. We appreciate your suggestions and look forward to improving the paper accordingly.

---

### Official Review · Reviewer_Bbzq · 2025-06-29

**Clarity:** 2
**Significance:** 2
**Originality:** 2
**Rating:** 4
**Confidence:** 4

**Summary:**

The paper introduces a new architecture based on Conditional Neural Processes (CNPs), named Spectral Convolutional Conditional Neural Processes (SConvCNPs). This model replaces the standard local convolutions in the spatial domain with global convolutions in the spectral domain, inspired by Fourier Neural Operators (FNOs). The authors argue that this modification helps address the limitations of CNPs, particularly in capturing long-range dependencies and handling sparse or irregularly sampled data. The method is evaluated on several synthetic and real-world datasets, where SConvCNPs perform comparably or better than existing baselines.

**Questions:**

1.  There seems to be something wrong with Eq 4, where in the second term in the parentheses, the two $\psi_{e}(t, x_i)$'s in the top and bottom can be cancelled out? I would guess it shoud be $\frac{\sum \psi_e(t, x_i) y_i}{\sum \psi_e(t, x_i)}$ instead. Also, the citation of ConvCNP seems to be wrong, it should be J. Gordon et. al. .
2.  Line 122: What does $C_b(\mathbb{X}, \mathbb{Y})$ mean? And Eq. 5 is not clear to me, could you explain it?
3.  Line 145: "the operator $\Phi$ in Equation 4 ...": I believe it's Eq. 5? Is it a typo?
4.  Line 150: "replaces ... as defined in Equation 3": I could not see how to plug Eq. 3 into Eq. 4 or 5, could you explain more?
5.  Line 159: "Althought $\tilde{E}$ breaks strict translation equivariance ...": Does it mean then the contextual points must maintain a specific spatial order? If not, how would you deal with it?
6.  Line 165 - 168: I do not quite understand what do you mean by "keeping the window's range and resolution constant". Is the window's range the kernel size? Which resolution are you referring to?
7.  Line 216: Empty citation?
8.  I am curious about the performance of SConvCNF in high-dimensional data, e.g., images, as the neural operators have great potentials in resolution-free inference. It would be nice to see the applications in super-resolution or image inpainting tasks.

**Ethical Concerns:**

["NO or VERY MINOR ethics concerns only"]

**Final Justification:**

During the discussions, the authors provided additional clarifications on the main theoretical constructions and agreed to reorganize the text with more details, which should help improve the quality of the paper. Therefore, I have decided to raise my score to positive.

**Limitations:**

The authors did not discuss the potential social impact of the work in the submission.

**Paper Formatting Concerns:**

The paper does not contain any major formatting issues.

**Quality:**

3

**Strengths And Weaknesses:**

**Quality**

This is a generally sound paper that introduces a new CNP-based model, SConvCNP, designed to improve global contextual conditioning in Conditional Neural Processes. The model demonstrates improved performance in the presented experiments, supported by detailed metric evaluations and comparisons with related models. Sufficient details of implementation are provided and the used codebase is promised to be released.

However, the theoretical grounding of the proposed method is limited. For example, it is unclear how the spectral convolution in Eq. 3 is integrated into the overall CNP framework. In addition, the main motivation of the paper is to address global and irregular conditioning. Yet, the experiments do not clearly reflect these scenarios, making it difficult to assess whether the proposed modification effectively addresses the stated challenges.

**Clarity**

The paper is generally easy to follow. The contribution is clearly stated in the introduction, and the necessary preliminary background is reviewed in Section 2. The experimental section provides sufficient details, along with comprehensive benchmarking tables and clear, readable figures, followed by a review of related work.

However, as noted earlier, the theoretical presentation of SConvCNP in Section 3 is oversimplified. The core component $\tilde{E}[\mathcal{D}] = (E[\mathcal{D}], t)$ is not clearly explained and may cause confusion. Additionally, Section 3.1 lacks an explicit discussion of the limitations of the standard CNPs that the proposed method aims to address (see questions below).

**Significance**

This work makes a useful step toward improving CNPs, but it would benefit from further validation, both in theory and in practice.

**Originality**

While SConvCNP presents a new architectural design, its two main components—CNP and FNO—are already well-established. Therefore, the combination, though practical and effective, may be viewed as incrementally novel rather than conceptually new.

---

> ### Comment · Reviewer_RBsp · 2025-08-02
> **Missing rebuttal?**
>
> Did the authors mean to not provide a rebuttal to reviewer Bbzq? I thought that the review from Bbzq was insightful and would like to see how the authors address these comments.

---

> ### Author Response · Authors · 2025-08-02
>
> Dear Reviewers,
>
> We sincerely apologize for the confusion — our rebuttal was mistakenly submitted as a confidential comment to the Area Chair rather than as a public response to your review.
>
> To clarify and ensure our responses are visible, we are posting our intended rebuttal below:
>
> **1. Theoretical Clarity and Equation Integration**
>
> We appreciate the reviewer’s feedback on the need for clearer theoretical exposition. We will revise Section 3 to include a more detailed derivation of how spectral convolution is integrated into the ConvCNP framework.
>
> Specifically:
> - Equation 5 will be rewritten to explicitly show how the Fourier-based operator is applied to the functional embedding of the context set.
> - We will provide a complete description of the model pipeline and clarify how each equation connects to the CNP formulation.
>
> We also thank the reviewer for pointing out specific issues, which will be corrected as follows:
>
> - Equation 4: The reviewer is correct that the numerator and denominator should be independently computed and should not cancel out. We will revise the expression accordingly.
>
> - Line 122: The notation $C_b(\mathbb {X}, \mathbb{Y})$ refers to the space of bounded continuous functions from $\mathbb{X}$, to $\mathbb {Y}$. Equation 5 expresses the encoder $\varphi_e$ as  $\varphi_e [ D ] ( x )  = \Phi ( E [ \mathcal{D} ] ( x ) ) $ where $ E [ D ] ( \cdot ) $ is the functional embedding mapping observations to a continuous function and $ \Phi $ is an operator.
>
> - Line 145: This should reference Equation 5, not Equation 4. We will correct the typo.
>
> - Line 150: To clarify how Equation 3 is used, we will add:
>   $$ ( \mathcal{L} [ E [ D ] ] ) ( x ) = \sigma ( W_r \cdot E [ D ] ( x ) + \mathcal{ F } ^{-1} [W_c \cdot \mathcal{ F } [ E [ D ] ] ( \omega ) ] ( x ) )  $$
>   where $ \mathcal{ F } $ and $ \mathcal{ F } ^ { -1 }$ denote the Fourier transform and its inverse, and $W_c$, $W_r$ are learnable parameters.
>
> **2. Translation Equivariance**
>
> Line 159: We clarify the notion of translation equivariance following Gordon et al. [Gordon et al., 2019]. Let
> $$ T_{ \tau } ( D ) = \\{ ( x + \tau , y ) | ( x , y ) \in D \\} $$,
> $ T'_{\tau} \varphi_e [ D ] ( x ) = \varphi_e [ D ] ( x - \tau ) $$
>
> Equivariance holds if:
> $\varphi_e [ T_{ \tau } ( D ) ] ( x ) = T'_{ \tau } \varphi_e [ D ] ( x ) $
>
> This requires both the embedding $E [ D ] $ and the operator $ \Phi $ to be equivariant. Adding positional encodings to $ E [ D ] $ (if not it's not based on relative distance) breaks this property, even if $\Phi$ itself remains shift-invariant.
>
> **3. Grid Resolution and Windowing (Lines 165–168)**
>
> By "keeping the window’s range and resolution constant," we mean that we evaluate $ E [ D ] $ on a fixed, uniform grid regardless of the number or location of context points.
> For example, in the 1D regression experiments, we use a grid over [-3, 3] with resolution 1/63.
>
> **4. Clarifications and Typos**
>
> - Line 216: Thank you—this was an incomplete citation and will be corrected.
>
> **5. Evaluation on Image Data**
>
> We included an image inpainting task using the Describable Textures Dataset (DTD, [Cimpoi et al., 2014]). This experiment demonstrates the model’s performance in a 2D real-world setting. The log-likelihood results are:
>
> - CNP: 0.66 ± 0.00
> - TNP: 1.70 ± 0.01
> - ConvCNP: 1.68 ± 0.05
> - SConvCNP: 1.74 ± 0.00
>
> **6. Societal Impact Discussion**
>
> We acknowledge the omission of an explicit societal impact statement. While our work is primarily methodological, we will include a discussion in the final version addressing fairness, potential misuse, and considerations for responsible deployment.
>
> **Conclusion**
>
> We thank the reviewer again for their detailed and thoughtful review. We will revise the paper to improve theoretical clarity, correct technical and typographical issues, and better highlight the model’s capabilities. We hope these improvements will address the reviewer’s concerns and strengthen the overall contribution.
>
> **Refrences**
> - Gordon, Jonathan, et al. "Convolutional conditional neural processes." arXiv preprint arXiv:1910.13556 (2019).
> - Cimpoi, Mircea, et al. "Describing textures in the wild." Proceedings of the IEEE conference on computer vision and pattern recognition. 2014.

---

> ### Comment · Reviewer_Bbzq · 2025-08-03
>
> I appreciate the authors’ replies and reviewer RBsp’s kind reminder. I think the replies address some of my questions, though two points still remain unclear to me:
>
> 1.Clarification of Equation 3
> To confirm: the role of $\mathcal{L}$ here is to replace $\mathrm{CNN}$ in Equation 3 of J. Gordon et al. since both operate on the embedding $E$, correct? If so, I suggest including the original formulation from J. Gordon in Section 3, so that readers unfamiliar with ConvCNP can quickly understand the connection.
>
> 2.Translation equivariance
> If I understand correctly, in the sequential data case, if translation equivariance holds, it means that the query point $x$ can be shifted anywhere (possibly within the contextual history) while the prediction $y$ remains unchanged. This seems counterintuitive to me, as the prediction would no longer appear to depend on the query point’s position. Would this not disrupt the sequential structure of the data? I am wondering whether this equivariance requirement is truly necessary for sequential prediction tasks.

---

> > ### Author Response · Authors · 2025-08-05
> >
> > We sincerely thank the reviewer for their follow-up and helpful suggestions. We address both remaining points below:
> >
> > 1.**Clarification of Equation 3**: Yes, that is correct: the operator $\mathcal{L}$ in the formulation serves the same role as the residual convolutional block in CNNs. We agree that including the original ConvCNP formulation would make the connection clearer for readers unfamiliar with that work, and we will incorporate a concise recap of the ConvCNP architecture in Section 3 for clarity.
> >
> > 2.**Translation equivariance**: This is a subtle but important point, and we appreciate the opportunity to clarify. When translation equivariance holds, predictions depend on the *relative positions* between the query point and the context points—not on their absolute locations [1]. This means that if *both* the query and the context points are shifted by the same amount, the prediction remains unchanged.To make this concrete, let the context set be  $ \mathcal{D} = \{ (x_i ^ { ( c ) } , y_i^{ ( c ) } ) \}_{ i = 1 } ^ { N_c } $
> > and let $ \varphi_e [ \mathcal{ D } ] $ be the functional embedding discretized over a grid $ \mathcal{ G } $.  Then, the prediction at a query point $ x^{(q)} $ is
> >
> > $$ \theta = \sum_{ t \in \mathcal{G} } \varphi_e [ \mathcal{ D } ] ( t )  \psi_d ( x^{ ( q ) }  - t). $$  Now consider a shifted query $\hat{x}^{(q)} = x^{(q)} - \tau$. The prediction becomes:
> >
> >  $$ \hat{ \theta } = \sum_{ t \in \mathcal{G} } \varphi_e [  \mathcal{ D } ] ( t )  \psi_d ( \hat{ x }^{ ( q ) } - t ) = \sum_{ t  \in \mathcal{ G } } \varphi_e [ \mathcal{ D } ] ( t )  \psi_d ( x^{ ( q ) } - ( \tau + t ) ) $$
> >
> > Letting $ \hat{ t } = t + \tau $ and $ \hat{ \mathcal{ G } } = \\{ t + \tau \mid t \in \mathcal{ G } \\} $, we rewrite this as:
> >
> >  $$ \hat{ \theta } = \sum_{ \hat{ t }  \in \hat{ \mathcal{ G } } } \varphi_e [ \mathcal{ D } ]( \hat{ t } - \tau ) \psi_d ( x^{ ( q ) } - \hat{ t } ) = \sum_{ \hat{ t }  \in \hat{ \mathcal{ G } } } \varphi_e [ T_{ \tau } \mathcal{ D } ]( \hat{ t }  ) \psi_d ( x^{ ( q ) } - \hat{ t } ) $$
> >
> > where the last equality holds since $ \varphi_e $ is translation equivariant. Inspecting the expressions of $ \theta $ and $ \hat{ \theta } $ shows that shifting the query point while holding the context fixed does not necessarily lead to the same predictions. Translation equivariance ensures consistent predictions only under joint shifts of both query and context, preserving relative geometry.
> >
> > In the sequential setting, where the position of events carries semantic meaning (e.g., temporal ordering), strict translation equivariance may not be desirable as it implies stationarity. In such cases, models may benefit from positional encoding schemes that preserve absolute position information, even though this breaks equivariance.
> > We emphasize that translation equivariance is not a central assumption or contribution of our method; rather, we discuss it to illustrate how operator learning techniques can be applied to functional data more broadly, beyond their traditional use in PDE modeling.
> >
> > We hope this clarification resolves the reviewer’s concerns and are happy to incorporate further elaboration if needed.
> >
> > [1]. Ashman, Matthew, et al. "Translation equivariant transformer neural processes." arXiv preprint arXiv:2406.12409 (2024).

---

> > > ### Comment · Reviewer_Bbzq · 2025-08-05
> > >
> > > I appreciate the authors’ further clarification and believe they have addressed my remaining concerns. With the potential revisions based on all the discussions, I expect the quality of the paper will improve. Therefore, I have decided to increase my score.

---

> > > > ### Author Response · Authors · 2025-08-06
> > > >
> > > > Thank you for your thoughtful feedback and for taking the time to reevaluate your score. We value your suggestions and are committed to incorporating them to strengthen the paper.

---

### Official Review · Reviewer_RBsp · 2025-06-30

**Clarity:** 1
**Significance:** 3
**Originality:** 3
**Rating:** 3
**Confidence:** 4

**Summary:**

This paper introduces Spectral Convolutional Conditional Neural Processes (SConvCNPs).
- This model builds on Convolutional Neural Processes (ConvCNP) in which the representation of conditional data points is translated to function space.
- The SConvCNP replaces the CNN in the ConvCNP with a Fourier Neural Operator (FNO) which the authors claim improves the ability of the model to capture long range dependancies.
- They benchmark their method on various 1D meta-learning regression tasks.

**Questions:**

- Can you clarify why the TNP figures for the square wave  look terrible, while the performance reported in Table 1 indicates equal performance to SConvCNP?
- Can you comment on your choice of synthetic functions from the perspective of bandwidth limited functions? The sawtooth and square wave are where your method performs better than the alternatives, but both of these have FTs which drop with 1/frequency, which I believe is a much slower decay than the GP examples. This seems to contradict your earlier discussion of frequency cutoff in line 63-65, so I am interested in your interpretation of the results.
- Can you comment on how your proposed method scales with the number of input dimensions?
- In line 98 does "complex" refer to the weights being a complex number (as is commonly used in FFTs), or just a statement that the weights can have arbitrary form? In line 102 you mention $W_r$ as being the real weights, making me thing $W_c$ are complex numbers, but I don't see why you would need complex numbers from this application. Additionally  $W_r$ appears to not be used.
- Can you explicitly define the following symbols:
	- $C_b$ on line 122?
	- $\theta$ on line 125? I assume that this is the mean and variance of the prediction, but would like this confirmed.
	- $t$ in equation 4? I assume this should be $t \in \mathbb{X}$.

**Ethical Concerns:**

["NO or VERY MINOR ethics concerns only"]

**Final Justification:**

I think that the research is interesting, and think that the idea is certainly interesting enough to merit publication. However, I feel that the authors have some issues with their evaluation which prevent me from recommending acceptance.

The experimental are carried out with their own method having far more parameters than alternatives, making it hard to establish if the improved performance is due to the architecture, or just having a larger model.

Additionally, the lack of ablation studies of their method, especially with respect to number of frequencies in the FNO,  make it hard to evaluate which features of their model are important.

**Limitations:**

The authors do briefly discuss the limitation of thier method in section 3.1. I would like a discussion of how this method scales in input dimension, as mentioned in my question, as I believe that this method would scale poorly with that.

**Paper Formatting Concerns:**

- Incorrect  bibtex entry on line 216
- Inconsistent number of significant figures in Table 1
- Bold should be used to indicate the best performance in each experiment in the tables.

**Quality:**

1

**Strengths And Weaknesses:**

Strengths
- The approach combines Neural Operators and Neural Processes, which is an exciting approach which I believe is currently under-explored. I thought this was a really interesting approach and was a paper which I enjoyed reading and thinking about the ideas they presented here.
- The results they present look promising, although I do have some issues with the current benchmarking which I mention below.

Weaknesses
- The referencing in this paper was sloppy, resulting in  concerning  errors (missing .bib entry on line 216, incorrect citation for ConvCNPs line 118, incorrectly referencing both Garnelo2018a in line 104).
- I would say that the current selection of benchmarks is weak:
	- The authors restrict their experimentation to only be one-dimensional examples. My understanding of this method is that it should generalise to higher dimensions than this but it is not explored.
	- The authors use only one real-world data example, with section 4.1 and 4.2 both being synthetic.
	- The authors do not provide the code. I am very surprised by the poor performance of the TNP in figure 1 (see also the question on this), and would have appreciated the ability to check the code implementation.
- The introduction of the model in Section 3 is missing details. I would have liked a restatement of the method, with the proposed changes, in an equation. At a minimum the writing needs to be tightened up significantly. For example, the authors refer to "$\Phi$ in equation 4" when $\Phi$ does not appear in equation 4, but does appear in equation 5.
- (Minor point.) I would disagree with the framing of the problem NPs are solving as "learning the covariance structure implicitly" on line 38, as my understanding is that NPs are actually capable of learning more complex dependencies than just covariances.

---

> ### Author Rebuttal · Authors · 2025-07-31
>
> We sincerely thank the reviewer for the thoughtful and constructive feedback. We appreciate your recognition of the novelty in combining neural operators with neural processes, and we are pleased that you found the ideas in our paper engaging. Below, we address each of your comments in detail.
>
> **(1) Referencing and Citation Errors**
>
> Thank you for identifying these issues. We will correct:
> - The missing `.bib` entry on line 216
> - The incorrect citation for ConvCNPs on line 118
> - The ambiguous reference to Garnelo et al. [2018a]
>
> These will be resolved in the revised manuscript to ensure accurate and consistent referencing.
>
> **(2) Benchmark Scope and Dimensionality**
>
> We agree that extending to higher-dimensional tasks is important. However, scaling ConvCNPs—and by extension, SConvCNPs—is challenging unless the data lies on a grid or one invests heavily in compute. This is due to the need for a fine discretization of the input domain before applying convolutions, as discussed in [Ashman et al., 2024].
>
> For example, in our synthetic 1D tasks, the domain [-3, 3] is discretized at 64 points/unit, yielding 384 grid points. Extending this to 2D with similar resolution leads to 384^2 points, which quickly becomes computationally prohibitive. Moreover, this resolution is often necessary for good performance.
>
> That said, we did include a real-world 2D image inpainting task using the Describable Textures Dataset [Cimpoi et al., 2014], demonstrating the model's performance on 2D real-world data. The loglikelihood results are:
>
> - CNP: 0.66 ± 0.00
> - TNP: 1.70 ± 0.01
> - ConvCNP: 1.68 ± 0.05
> - SConvCNP: 1.74 ± 0.00
>
> **(3) Code Availability and TNP Performance**
>
> We used the official codebase from [Ashman et al., 2024] to train all transformer-based models (TNP, TETNP), adhering strictly to their training and hyperparameter protocols. The poor performance of TNP in Figure 1 appears to result from the *spectral bias* of transformers—i.e., their tendency to prioritize smooth, low-frequency signals.
>
> This is consistent with findings in [Nguyen and Grover, 2022], where TNP underperforms even vanilla CNPs on GP data with periodic kernels. Additional studies (e.g., [Vasudeva et al., 2024]) support this, highlighting transformers' difficulty in modeling high-frequency components. We retrained TNP and TETNP with larger capacities, and the performance remained unchanged, reinforcing this hypothesis.
>
> **(4) Inconsistency Between Figure 1 and Table 1 for TNP on Square Wave**
>
> Thank you for catching this error. Upon rechecking, we found that the TNP results for the square wave in Table 1 were inadvertently copied from another experiment. We have re-run the benchmark and confirmed that the numerical results match the poor qualitative performance in Figure 1. For TNP on squarewave, the log-likelihoods averaged over 4 seeds was $-1.39 \pm 0.00$. This correction further highlights the limitations of TNP on high-frequency signals.
>
> **(5) Missing Equation and Notation Issues in Section 3**
>
> We will revise Section 3 to provide:
>
> - A clearer restatement of our method with explicit equations comparing SConvCNP and ConvCNP
> - Corrections to referencing errors (e.g., "Equation 4" should be "Equation 5")
> - Clarified notations, including:
>   - Line 122: C_b(X, Y) is the space of bounded continuous functions from X to Y
>   - Line 125: θ denotes parameters of the predictive distribution (i.e., mean and variance for Gaussian outputs)
>   - Equation 4: t ∈ X (analogous to x); we will unify notation for clarity
>
> **(6) Line 98 – "Complex Weights"**
>
> In line 98, "complex" refers to complex-valued weights, which are standard in frequency-domain methods using FFT. These allow flexibility in modeling kernels without assuming real-valued spectra. By contrast, W_r (line 102) are real-valued weights used for pointwise channel mixing, akin to residual projections in CNNs or transformers. We will clarify these design choices in the revised text.
>
> **(7) Function Choice and Bandwidth in Synthetic Benchmarks**
>
> We appreciate this nuanced observation. Indeed, sawtooth and square waves have slower spectral decay, and our method’s strong performance on them stems from two key factors:
>
> 1. **Periodic alignment**: The FFT assumes periodicity, which aligns well with these synthetic functions, enabling accurate spectral representations
> 2. **Energy distribution**: On a fixed grid, the FFT returns fixed frequency bins. For a GP with a Matérn-5/2 kernel, most energy is concentrated in low frequencies, while periodic functions like sawtooth and square waves spread the same energy across a wider band. As a result, the set of these fixed FFT modes is more likely to capture informative components for periodic functions
>
> We will include a discussion of these points to clarify how function class influences performance.
>
> **(8) Scaling with Input Dimension**
>
> As noted, SConvCNP inherits ConvCNP’s scalability challenges due to the discretization bottleneck. While FNOs improve parameter efficiency and receptive field size compared to CNNs, the cost of discretizing the entire input space (especially in 2D or 3D) still grows exponentially with dimension.
>
> **(9) Formatting and Presentation**
>
> We appreciate the detailed formatting feedback. We will:
>
> - Fix the `.bib` entry on line 216
> - Ensure consistent significant figures in Table 1
> - Bold the best performance values in all result tables
>
> **Refrences**
> - Cimpoi, Mircea, et al. "Describing textures in the wild." Proceedings of the IEEE conference on computer vision and pattern recognition. 2014.
> - Gordon, Jonathan, et al. "Convolutional conditional neural processes." arXiv preprint arXiv:1910.13556 (2019).
> - Nguyen, Tung, and Aditya Grover. "Transformer neural processes: Uncertainty-aware meta learning via sequence modeling." arXiv preprint arXiv:2207.04179 (2022).
> - Ashman, Matthew, et al. "Translation equivariant transformer neural processes." arXiv preprint arXiv:2406.12409 (2024).
> - Vasudeva, Bhavya, et al. "Transformers Learn Low Sensitivity Functions: Investigations and Implications." arXiv preprint arXiv:2403.06925 (2024).

---

> > ### Comment · Reviewer_RBsp · 2025-08-04
> >
> > I thank the authors for their response, especially their interesting discussion of the spectral biases of TNPs. I think that the research is a compelling idea, however I still have concerns:
> >
> >
> > 1. The experimental setup
> >
> > Details given in the DTD example given in the rebuttal  seem to be missing both the RMSE metric  the TETNP which seems to be on of the strongest baseline comparisons. Along with the very limited description of the setup given in the rebuttal, I don't think that I can "give full credit" for the additional experiment.
> >
> > Additionally on rereading the work, I found it hard to convince myself fully that the suggested approach was certainly better than the alternatives as I believe the authors have not provided details like  the number of parameters in each of the models.
> >
> >
> >
> > 2. Insufficient exploration of performance of their method with respect to features the power spectrum
> >
> >
> > While the additional description of the low frequency biases of TNPs in the rebuttal was very interesting, it seems to fly in the face of the stated motivation for introducing the FNO in the first place, which in the introduction they say is because low frequencies carry most of the information of the signal. Their method clearly seems to perform well in domains with lots of high frequency information. In point (7) of their rebuttal the authors have not really addressed this, where they claim better performance due to periodicity. The FFT assumes periodicity outside of the range of the signal, it is not obvious to me why they expect periodicity inside of the observed range of inputs to result in better performance for their method.
> >
> >
> > Furthermore, the predictions which they present for the Sawtooth and Square Waves seem to have high uncertainties at the boundaries in a way which is highly reminiscent of Gibbs phenomena, which would be expected from performing Fourier transforms with insufficiently many frequencies included. I believe the authors have not provided any analysis of the sensitivity of their method to the number of frequencies, which I would say is a highly salient piece of information for anyone hoping to use their method.
> >
> >
> > As a the authors talk about it in their introduction I would say that this is a fundamental (and interesting) area to explore with their method, but the authors do not take advantage of this to illustrate improvements compared to other NP setups.

---

> > > ### Author Response · Authors · 2025-08-06
> > > **The experimental setup**
> > >
> > > We appreciate the reviewer’s feedback and the opportunity to clarify our experimental setup on the DTD benchmark. To maintain brevity in the rebuttal, we omitted full details, but we now provide them below for completeness:
> > >
> > > **(1) Dataset**: We used the default DTD dataset split, consisting of 1,880 images each for training, validation, and testing. All images have 3 channels. Image dimensions range from 231–778 in height and 271–900 in width. During both training and evaluation, we randomly crop a 192×192 patch from each image and resize it to 64×64. This introduces more variation per sample without significantly increasing the number of pixels. Random cropping is performed on-the-fly during training, while fixed random crops (shared across models and seeds) are used for evaluation. Pixel coordinates are rescaled to the range [-1, 1], and pixel intensities are normalized to [0, 1].
> > >
> > > **(2) Training**: All models are trained for 500 epochs using AdamW with a learning rate of $ 1 \times 10^{-4} $, gradient clipping at 0.5, and a batch size of 16. Each model is trained using 4 random seeds. For each training and evaluation sample, the number of context points is uniformly sampled from [5, 2048], with the remaining points used as targets.
> > >
> > > **(3) Model details**:
> > >
> > > - *Output parameterization*: For all models, the predicted mean is passed through a sigmoid activation to ensure outputs lie in [0, 1]. The predicted scale is passed through a softplus activation, scaled by 0.99, and increased with a minimum noise floor of 0.01.
> > >
> > > - *CNP*: The CNP architecture mirrors that of prior experiments and contains approximately 1.3 million parameters.
> > >
> > > - *TNP*: The TNP architecture follows the setup from earlier experiments, with the main difference being the use of 6 attention layers, each with 12 heads of dimension 16. All intermediate embeddings have a dimension of 512, resulting in approximately 6.6 million parameters.
> > >
> > > - *ConvCNP*: We use a standard ConvCNP architecture with 5 residual CNN blocks. Each block has 128 input/output channels and a kernel size of 11, resulting in approximately 12.2 million parameters.
> > >
> > > - *SConvCNP*: We adopt a U-shaped architecture with 7 blocks of depthwise separable convolutions. The first 4 blocks (pre-U) each have 512 input/output channels and use 32, 8, 2, and 2 Fourier modes, respectively. The second and third blocks operate on inputs downsampled by factors of 4 and 16, respectively, along each spatial dimension. The fourth block maintains the same resolution. In the final 3 blocks (post-U), the input/output channels are 1024/512. The fifth block receives concatenated outputs from the third and fourth blocks and upsamples by a factor of 4. The sixth block similarly upsamples the concatenated outputs of the fifth and second blocks. Finally, the seventh block processes the concatenation of the sixth block and the first block, without changing spatial resolution. The total number of parameters is approximately 14.2 million.
> > >
> > >
> > > **(4)**: We did not include TETNP in this benchmark due to its inefficiency with large numbers of context points and frequent out-of-memory issues.
> > >
> > >
> > > **(5) Parameter count**:  For the other benchmarks reported in the paper, the model architectures are kept consistent unless the input-output dimensionality differs, which has a negligible effect on the total number of parameters. Below, we report the number of parameters for each model on the synthetic benchmarks:
> > >    - CNP: 1.3 million
> > >    - AttCNP: 3 million
> > >    - TNP: 2.2 million
> > >    - TETNP: 2.7 million
> > >    - ConvCNP: 3.8 million
> > >    - SConvCNP: 4.7 million

---

> > > ### Author Response · Authors · 2025-08-06
> > > **Insufficient exploration of performance of their method with respect to features the power spectrum**
> > >
> > > **(1) Our claim**: We would like to clarify that we do not claim that our model is ``certainly better than the alternatives.'' Each model variant has its own strengths and limitations. For example, transformer-based models often achieve strong performance across a wide range of benchmarks, but their quadratic complexity in the number of data points makes them computationally demanding. Moreover, as observed in the synthetic benchmarks, they can suffer from spectral bias. In contrast, ConvCNPs are computationally efficient, with linear complexity in the number of data points, but they quickly become intractable in higher-dimensional domains.
> > >
> > > Our goal in this work is not to propose a universally superior variant. Rather, we aim to highlight the promise of the growing operator learning literature -- traditionally focused on PDEs -- for broader applications involving functional data. Our proposed approach explores how ideas from this domain can benefit neural process-style models.
> > >
> > > **(2) Motivation vs. Results**: We clarify that our motivation for operating in the frequency domain and restricting to a subset of Fourier modes is not inherently at odds with the improved performance observed on signals like sawtooth and square waves.
> > >
> > > - First, a desirable design choice is to expand the effective receptive field of convolutional layers in a computationally tractable way. In the spatial domain, doing so would require extremely large kernels—e.g., a kernel of size 384 for a domain of length 6 discretized into 384 points. In contrast, in the frequency domain, we can approximate global convolutions efficiently by learning filters over a much smaller number of modes (e.g., 32), significantly reducing the number of parameters and enabling global context aggregation.
> > >
> > > - Second, while our motivation leverages the observation that natural signals often concentrate energy in low frequencies, we acknowledge that the terms low frequency and high frequency are relative and context-dependent. For example, what we refer to as the “lower frequency region” does not imply only the very lowest frequencies in absolute terms, but rather a tractable subset of the spectrum that captures a large portion of the signal’s energy or structure. In practice, this region may include moderately high frequencies.
> > >
> > > - In this sense, our use of the term “low-frequency regime” refers to a restricted band of frequencies that is small relative to the full spatial resolution, not necessarily to signals with decaying spectra of a certain order. Even for signals with non-negligible high-frequency content, a truncated Fourier representation can retain enough structural information to perform well, as is the case for sawtooth and square waves.
> > >
> > > **(3) Gibbs phenomena**: We appreciate the reviewer’s observation regarding the elevated uncertainty near discontinuities in the Sawtooth and Square Wave experiments and their resemblance to Gibbs phenomena. While our current work does not delve into the analysis of this effect, it is a very interesting point and represents a valuable direction for future investigation.
> > >
> > > **(4) Number of modes**: Regarding the sensitivity to the number of Fourier modes, we agree that an ablation study would be informative, particularly for practitioners seeking to balance accuracy and efficiency. We will include an ablation study on the number of frequency modes in the revised version of the paper to provide clearer guidance for practical use.

---

> > > > ### Comment · Reviewer_RBsp · 2025-08-06
> > > >
> > > > I thank the authors for the continued engagement. The inclusion of the additional information given here in the paper would reduce my major concerns. Some additional clarifications/comments are below.
> > > >
> > > > **The experimental setup**
> > > >
> > > > Including the discussion in points (1) - (3) addresses most of my concerns around not being able to fully credit the experiment in my review.
> > > >
> > > > > (4)
> > > >
> > > > I would be happy with this reason for not including the TETNP. The missing RMSE metric is still a minor issue, as it is present in all of the other experiments, but the LogLik is more important.
> > > >
> > > >
> > > > > (5) Parameter count
> > > >
> > > > I think that the discrepancy in the number of parameters does weaken the results presented in this paper. With the current setup, I can't feel confident about the extent to which improved performance versus what is due to the novel Neural Process architecture and what is due to  simply fitting a bigger model.
> > > >
> > > >
> > > >  **Insufficient exploration of performance of their method with respect to features the power spectrum**
> > > > >  (1) Our claim
> > > >
> > > > I would like to clarify that I agree with the authors - it should be expected that no model is universally the best. I included my comment as if the experiments presented had achieved a clear SOTA performance it would have been a very strong argument for publication, but a clear and universal SOTA performance would definitely not be required for me to recommend acceptance.
> > > >
> > > > The better comment, which I should have made in the last exchange, is that the authors could do a better job of explaining under what circumstances their method should be better than alternatives and tailored both their writing and experiments to emphasise this. I think that the discussion which they have provided in their last round of comments significantly clarifies their motivation for introducing this architecture.
> > > >
> > > > > (4) Number of modes
> > > >
> > > > The inclusion of this ablation study would be an excellent addition to the paper. However, I am reluctant to consider this in my review unless the results of the study can be presented here. I am happy to revise this approach if the Area Chair thinks this is inappropriate.

---

> ### Author Response · Authors · 2025-08-07
>
> We sincerely thank the reviewer for the thoughtful feedback and continued, productive engagement throughout the discussion phase. We are glad to hear that the clarifications provided have addressed many of your concerns, particularly regarding the experimental setup and our motivation.
>
> Regarding the remaining issues:
>
> - Ablation on the number of modes: We fully agree that this study would significantly strengthen the paper. We are currently running the corresponding experiments, but due to computational constraints and the time remaining in the discussion phase (less than 24 hours), we are afraid we may not be able to share the results before the deadline. Nonetheless, we are committed to including this ablation in the final version of the paper.
> - Parameter count and larger baselines: We are also working on evaluating larger baselines that are better matched in parameter count and will make sure to report these results in the final version as well.
>
> We greatly appreciate your constructive suggestions throughout the review process, which have contributed meaningfully to improving the quality and clarity of our work. If there are any remaining concerns or suggestions beyond those already discussed, we would be more than happy to address them.

---

### Official Review · Reviewer_jGd3 · 2025-07-12

**Clarity:** 3
**Significance:** 3
**Originality:** 3
**Rating:** 4
**Confidence:** 3

**Summary:**

Convolutional Conditional Neural Processes are modified by replacing the standard convolution with a spectral convolution. The authors show that this is advantagous for modeling cyclic and long-range dependencies by plugging Frequency Neural Operator (FNO)-style spectral convolution layers into the ConvCNP architecture, replacing the original local, spatial-domain convolutions.
Spectral convolution enables the model to learn global, task-adaptive kernels in the frequency domain.
The authors introduce truncated frequency filtering for efficiency and regularization.
Positional encodings plus a sliding-window FFT mitigate discretisation artefacts.

The authors evaluate the model on synthetic data and real-world data.
Comparison methods are CNP, Attentive CNP, TNP, TETNP, and ConvCNP. The results show that the spectral convolution is able to capture long-range dependencies, especially for cyclic dependencies.

2 minor errors:
- LargeST [?] (reference is missing.)
- This frequency truncation is the key to *reduces* the computational complexity.

**Questions:**

The comparison of predictive performance reported in Table 1 and Table 2 and Table 3shows that the proposed method provides results that are either statistically equivalent or better than the comparison methods. However, looking at the numbers, the paper reports numerical values with extremely high precision in significant digits and reports very small (often exactly 0) standard deviation on RMSEs and log-likelihoods. Given that 4 seeds where used for generation in each experiment, the reported standard deviations may be underestimated, or the generated data is (close to) deterministic.

What is the runtime of the method vs. dataset size in comparison to other related methods?

How do the discussed limitations and provided fixes effect robustness of the method and the results?

**Ethical Concerns:**

["NO or VERY MINOR ethics concerns only"]

**Final Justification:**

I thank the authors for taking on some of my recommendations, promising ablation studies and runtime analyses. As only limited results have been shown during the rebuttal phase, I these did not significantly change my review.

**Limitations:**

Given that there are a few design experiments discueed in the paper (discretisation, sliding window, positional encodings, etc.), I would have liked to see a more detailed analysis of the design choices and their impact on robustness of the models and the results, e.g. using a more detailed ablation study.

**Paper Formatting Concerns:**

The formatting looks good.

**Quality:**

3

**Strengths And Weaknesses:**

Incorporating spectral convolutions into the ConvCNP framework is conceptually elegant and practically valuable, as it enables global receptive fields without increasing kernel size.
However, while effective, it is a relatively targeted architectural change, rather than a broader theoretical or methodological contribution.

The experimental results show that the spectral convolution is able to capture long-range dependencies, especially for cyclic dependencies on par or better than the comparison methods.

Lack of assessment of uncertainty calibration. The fact that the model outputs uncertainty estimates are a key benefit compared to simpler models.

Given that there are a few design experiments discueed in the paper (discretisation, sliding window, positional encodings, etc.), I would have liked to see a more detailed analysis of the design choices and their impact on robustness of the models and the results, e.g. using a more detailed ablation study.

Runtime experiments are not provided in the paper, especially as the method is motivated (amongst other things) to avoid cubic runtime complexity of GPs.
In fact, GPs are missing from the comparison methods comparison. As these are the most common methods for probabilistic modeling of functions, it would have been interesting to see how the proposed method compares to them.

---

> ### Author Rebuttal · Authors · 2025-07-31
>
> We thank the reviewer for the thoughtful and constructive feedback. We are pleased that you found our spectral convolutional modification "conceptually elegant and practically valuable," and appreciate your recognition of the model’s strong empirical performance on long-range and cyclic dependencies. We address your comments below.
>
> **(1) Uncertainty Calibration**
>
> We agree that uncertainty quantification is a key strength of neural processes (NPs). While we acknowledge that additional metrics—such as calibration error—could provide a more comprehensive assessment, we followed standard practice in the NP literature [Gordon et al., 2019; Bruinsma et al., 2023; Lee et al., 2023; Ashman et al., 2024] by using log-likelihood as our primary evaluation metric. As a proper scoring rule, log-likelihood directly measures the quality of probabilistic predictions and remains a widely accepted benchmark for assessing uncertainty in generative models [Lakshminarayanan et al., 2017].
>
> **(2) Ablation Studies on Design Choices**
>
> Thank you for the suggestion. Our discretization and hyperparameter settings follow prior work [Bruinsma et al., 2023], and positional encodings were adopted from UNO [Rahman et al., 2022] using Cartesian grid coordinates. While we did not employ the sliding window in our current experiments (as our domains were large enough), we mentioned it as a potential extension for future applications (e.g., forecasting). That said, we will expand the ablation study in the revision to examine the effects of discretization density and positional encoding.
>
> **(3) Runtime and Efficiency Analysis**
>
> We agree that runtime comparisons are crucial, especially given that one of our core motivations is the efficiency of global convolution. Our method scales linearly with the size of the context and target sets, in contrast to attention-based models and Gaussian Processes, which exhibit quadratic and cubic complexity, respectively. While our approach employs the FFT as part of the FNO architecture—resulting in a complexity of $ \mathcal{O} ( | \mathcal{ G } | \log | \mathcal{G} | ) $ for processing the functional embedding over the grid $ \mathcal{G} $—this is only slightly higher than the $ \mathcal{O} ( |\mathcal{G}| ) $ cost of spatial convolutions used in ConvCNPs. In practice, however, the FFT implementation in PyTorch is highly optimized, introducing minimal computational overhead. To support our claims, we will include runtime vs. dataset size plots, along with benchmarks of wall-clock time and memory usage.
>
> Average forward runtime for an image completion task:
> - SConvCNP: 0.042 s
> - ConvCNP: 0.022 s
> - TNP: 0.07 s
> - CNP: 0.005 s
>
> **(4) Comparison to Gaussian Processes (GPs)**
>
> Thank you for pointing this out. We focused on NP baselines to highlight architectural advancements within the same framework. Comparing GPs to NPs is not straightforward: GPs are typically fit independently per task (with no inter-task learning), while NPs amortize across a distribution of tasks [Kim et al., 2019]. Moreover, while GP kernels like Matern or periodic are compatible with some of our synthetic tasks, no analytic kernel is known to (e.g., sawtooth or square waves) such that the induced GP coincides with the used processes for generating data.
>
> **(5) Precision and Standard Deviation Reporting**
>
> You are right to question the very low standard deviations. While each experiment used 4 random seeds and multiple model inits, the small variance arises from: (1) deterministic model components, (2) large numbers of training tasks, and (3) a fixed test task set. This phenomenon is also observed in prior work [Gordon et al., 2019; Bruinsma et al., 2023].
>
> **(6) Corrections**
>
> Thank you for catching these:
> - "LargeST" will be corrected to "largest" or otherwise revised for clarity.
> - "reduces the computational complexity" will be revised to "reduce the computational complexity."
>
> **Summary**
>
> We thank the reviewer again for their constructive feedback. In response, we will:
> - Expand the ablation study on design components.
> - Include runtime and memory benchmarks.
>
> **Refrences**
>
> - Rahman, Md Ashiqur, Zachary E. Ross, and Kamyar Azizzadenesheli. "U-no: U-shaped neural operators." arXiv preprint arXiv:2204.11127 (2022).
>
> - Lakshminarayanan, Balaji, Alexander Pritzel, and Charles Blundell. "Simple and scalable predictive uncertainty estimation using deep ensembles." Advances in neural information processing systems 30 (2017).
>
> - Bruinsma, Wessel P., et al. "Autoregressive conditional neural processes." arXiv preprint arXiv:2303.14468 (2023).
>
> - Gordon, Jonathan, et al. "Convolutional conditional neural processes." arXiv preprint arXiv:1910.13556 (2019).
>
> - Ashman, Matthew, et al. "Translation equivariant transformer neural processes." arXiv preprint arXiv:2406.12409 (2024).
>
> - Lee, Hyungi, et al. "Martingale posterior neural processes." arXiv preprint arXiv:2304.09431 (2023).

---

> > ### Author Response · Authors · 2025-08-06
> > **Follow-up on Paper 26003 – Request for Feedback**
> >
> > Dear Reviewer jGd3,
> >
> > We understand you are likely managing multiple reviews, and we appreciate your time and efforts. As the discussion period is nearing its end (with less than 48 hours remaining), we would be very grateful if you could share any additional thoughts or feedback on our rebuttals. Your insights would be invaluable in helping us further improve the paper.
> >
> > Kind regards,
> > The authors of paper 26003

---

> > ### Comment · Reviewer_jGd3 · 2025-08-07
> >
> > Thank you for the response. While log-likelihood is a proper scoring rule and used frequently in the literature, its scale is often hard to interpret and compare between different models. Therefore, I recommend adding a measure that relates better to units of the data, such as  an expected MSE or similar.
> >
> > Thank you for promising an extension of the provided ablation studies. This will certainly improve the paper. As no results have been provided, I cannot assess the proposed ablation studies.
> >
> > Thank you for promising to include runtime results. This will help the reader better understand the dependencies between computational effort and model quality the models provide.

---

### Note · Authors · 2025-08-16

We conducted ablation studies to examine three key design choices: number of retained Fourier modes, discretization resolution, and inclusion of positional embedding (PE). We evaluated 1D functions drawn from a GP with a Matérn 5/2 kernel and on a sawtooth wave, to assess sensitivity across different function classes. In all experiments, we used a slightly modified U-shaped Neural Operator, consistent with our synthetic benchmarks, but with a reduced parameter count (3.1M parameters when 32 modes are retained). The training procedure followed the same protocol as in the benchmarks (4 seeds). For evaluation, we sampled 4096 tasks from the generative process. Reported numbers are log-likelihoods (higher is better).

The results can be summarized as follows:
- **Number of Fourier modes (m).** Increasing the number of modes substantially improves performance on the sawtooth wave, while yielding no noticeable change on the Matérn 5/2 kernel. This aligns with reviewer RBsp’s comment that the Fourier coefficients of a sawtooth wave
decay more slowly than those of Matérn 5/2 samples.
- **Discretization resolution (r spacing between consecutive samples of the functional embedding).** Finer discretization significantly benefits the sawtooth case but has little effect on Matérn 5/2. Again, this is consistent with the slower decay rate of the sawtooth Fourier spectrum, implying a higher Nyquist frequency and thus
requiring denser sampling of the functional embedding.
- **Positional encoding (PE).** Adding positional encoding further improves performance particularly on the sawtooth wave.


|         | m = 8         | m = 16        | m = 32        |
|------------------|---------------|---------------|---------------|
| Matérn 5/2       | -0.29 ± 0.00  | -0.29 ± 0.00  | -0.29 ± 0.00  |
| Sawtooth    | -0.14 ± 0.03  |  0.20 ± 0.02  |  0.80 ± 0.03  |


|         | r = 1/15      | r = 1/31      | r = 1/63      |
|------------------|---------------|---------------|---------------|
| Matérn 5/2       | -0.30 ± 0.00  | -0.29 ± 0.00  | -0.29 ± 0.00  |
| Sawtooth    |  0.03 ± 0.02  |  0.26 ± 0.08  |  0.20 ± 0.02  |


|         | With PE       | Without PE    |
|------------------|---------------|---------------|
| Matérn 5/2       | -0.29 ± 0.00  | -0.31 ± 0.00  |
| Sawtooth    |  0.80 ± 0.03  |  0.67 ± 0.02  |

We hope these ablation studies clarify the role of each design choice and address the remaining concerns raised by the reviewers.

---

### Decision · Program_Chairs · 2025-09-17

**Decision:**

Accept (poster)

**Comment:**

All reviewers found the core idea of combining Fourier Neural Operators (FNOs) with Convolutional Conditional Neural Processes (ConvCNPs) to be novel, interesting, and conceptually elegant. There was a consensus that the proposed method demonstrates strong performance, in particular in periodic signals and irregular signals (e.g. sawtooth).

The initial main concerns were about:
1. Experimental scope and validity: missing important ablation studies.
2. Model size differences in the comparison: The proposed model is larger and therefore it is not clear where the gain is coming from.
3. Clarity and errors: The description of the method was not detailed enough and some errors were found.

After these points were discussed with the authors, most reviewers were satisfied with the provided answers and additional results.

The final recommendations of the reviewers are: 3 reviewers recommend acceptance, and 1 reviewer is willing to accept the paper but is more reluctant due to the large amount of updates required after the discussion period.

After reading the reviews and the discussion, I believe this paper should be accepted, provided that the authors make all the updates mentioned in the discussion.